# Resolvin-D2 targets myogenic cells and improves muscle regeneration in Duchenne muscular dystrophy

Junio Dort[1,2], Zakaria Orfi[1,3], Paul Fabre [1,3], Thomas Molina [1,3], Talita C. Conte[1,3], Karine Greffard[4], Ornella Pellerito[1], Jean-François Bilodeau[4,5] & Nicolas A. Dumont [1,2✉]

Lack of dystrophin causes muscle degeneration, which is exacerbated by chronic inflammation and reduced regenerative capacity of muscle stem cells in Duchenne Muscular Dystrophy (DMD). To date, glucocorticoids remain the gold standard for the treatment of DMD. These drugs are able to slow down the progression of the disease and increase lifespan by dampening the chronic and excessive inflammatory process; however, they also have numerous harmful side effects that hamper their therapeutic potential. Here, we investigated Resolvin-D2 as a new therapeutic alternative having the potential to target multiple key features contributing to the disease progression. Our in vitro findings showed that Resolvin-D2 promotes the switch of macrophages toward their anti-inflammatory phenotype and increases their secretion of pro-myogenic factors. Moreover, Resolvin-D2 directly targets myogenic cells and promotes their differentiation and the expansion of the pool of myogenic progenitor cells leading to increased myogenesis. These effects are ablated when the receptor Gpr18 is knocked-out, knocked-down, or blocked by the pharmacological antagonist O-1918. Using different mouse models of DMD, we showed that Resolvin-D2 targets both inflammation and myogenesis leading to enhanced muscle function compared to glucocorticoids. Overall, this preclinical study has identified a new therapeutic approach that is more potent than the gold-standard treatment for DMD.

[1] CHU Sainte-Justine Research Center, Montreal, QC, Canada. [2] School of rehabilitation, Faculty of Medicine, Université de Montréal, Montreal, QC, Canada. [3] Department of pharmacology and physiology, Faculty of Medicine, Université de Montréal, Montreal, QC, Canada. [4] Endocrinology and Nephrology Unit, CHU de Québec-Laval University Research Center, Quebec city, QC, Canada. [5] Department of Medicine, Faculty of Medicine, Laval University, Quebec city, QC, Canada. ✉email: nicolas.dumont.1@umontreal.ca

Duchenne muscular dystrophy (DMD) is a severe disease that affects ~ 1 out of 4,000 boys worldwide and is characterized by progressive muscle wasting and weakness[1]. The disease is caused by X-linked recessive mutations in the DMD gene, which encodes for the dystrophin protein. The full-length dystrophin protein is constituted of an N-terminal domain that binds to the actin cytoskeleton, a C-terminal domain that connects to the dystroglycan complex, and a rod domain that contains numerous spectrin repeats acting as molecular shock absorbers during muscle contraction[2]. The absence of dystrophin triggers a complex cascade of events such as increased myofiber membrane permeability, calcium influx, reactive oxygen species (ROS) production, and cellular necrosis. Chronic muscle degeneration leads to the disorganized and persistent accumulation of inflammatory cells, which participates in disease progression[3]. Particularly, the presence of macrophages expressing high levels of TGF-β impairs muscle regeneration and promotes muscle fibrosis[4,5]. Moreover, muscle stem cells (MuSC), the engine of muscle repair, are also defective in DMD. In healthy regenerating muscle, activated MuSC enter cell cycle to become proliferating myoblasts that eventually exit the cell cycle to self-renew or to differentiate and fuse to form new muscle fibers[6,7]. Accumulating evidence indicates that the myogenesis capacity of dystrophin-deficient MuSC is impaired in DMD. Acute injury to dystrophic muscles leads to a deficit in MuSC activation and delayed/ incomplete regeneration[8–10]. In vitro characterization of myogenic cells revealed a reduction in cell proliferation, differentiation, and fusion[11–13]. The contribution of extrinsic factors (e.g., chronic inflammation) vs. intrinsic mechanisms to these defects still remains to be clearly defined; however, our previous work indicated that the lack of dystrophin in MuSC intrinsically affects their capacity to perform asymmetric cell division, a process during which one MuSC generates one self-renewing MuSC and one committed myoblast[14,15]. Overall, disease progression in DMD is multifactorial and is attributable to muscle fiber fragility, harmful inflammatory environment, and reduced myogenesis capacity of MuSC.

To date, there is no cure for DMD. From the hundreds of clinical trials tested for DMD, only glucocorticoids consistently demonstrated efficacy on the preservation of muscle function and ambulation; however, these effects are time-limited[16,17]. While the exact mechanism of action of glucocorticoids is not fully elucidated, evidence suggests that their therapeutic potential is mostly mediated through their anti-inflammatory properties[18]. Unfortunately, glucocorticoids have numerous detrimental side effects such as osteoporosis/fracture, cushingoid syndrome, anxiety, growth delay, among others[16,17]. More importantly, glucocorticoids are known to activate muscle protein degradation pathways (e.g., MuRF-1, Atrogin-1, FoxO1), inhibit protein synthesis pathways (e.g., Akt-1), and stimulate long-term muscle wasting[19]. Moreover, it was also shown that glucocorticoids impair myoblast proliferation and differentiation[20,21]. Therefore, the clinical outcome of increased muscle strength in DMD patients treated with glucocorticoids is mitigated by their harmful side effects, which could explain their transient efficacy. Thus, the medical need for treatment with higher therapeutic efficacy and fewer side effects than glucocorticoids is compelling.

Recently, a class of molecules named specialized pro-resolving mediators was shown to play a key role in the resolution of inflammation. These mediators originate from the conversion of omega-3 fatty acids (eicosapentaenoic acid or docosahexaenoic acid) by enzymes like 5-lipoxygenase and 15-lipoxygenase[22]. Particularly, Resolvins were shown to actively stimulate the resolution of inflammation by blocking neutrophil infiltration and promoting their apoptosis, stimulating the non-phlogistic phagocytosis by macrophages and their switch toward the anti-

inflammatory phenotype, decreasing ROS production, and inhibiting the expression of pro-inflammatory cytokines[23–25]. Many studies demonstrated the therapeutic potential of Resolvins for the treatment of immune-related conditions such as asthma[23], type 2 diabetes[24], sepsis[26], arthritis[27], among others[25]. Very few studies have investigated the role of specialized pro-resolving mediators in skeletal muscles. One study showed that Resolvin D2 (RvD2) expression peaks 5 days after hindlimb ischemia in skeletal muscle; and that treatment with RvD2 promoted tissue revascularization[28]. Another recent study showed that Resolvins peak between day 2 and day 4 after an acute injury induced by cardiotoxin or eccentric exercise[29]. In vivo treatment with RvD2 was shown to promote the transition of macrophages from Ly6C$^{hi}$ (pro-inflammatory macrophages) to Ly6C$^{lo}$ (anti-inflammatory macrophages), and to increase muscle mass and muscle force recovery after acute cardiotoxin injury[29]. Similar results were observed in regenerating mice (BaCl$_2$ injury), in which RvD1 administration dampened inflammation and increased muscle regeneration[30]. However, the therapeutic impact of RvD2 in a pathological condition affecting skeletal muscles such as DMD is unknown.

Here, we investigated the therapeutic potential of RvD2 compared to the standard-of-care treatment (glucocorticoids) for DMD. Using in vitro and in vivo approaches, we showed that RvD2 dampens inflammation and promotes the switch of macrophages toward their anti-inflammatory phenotype to a similar level to glucocorticoids. More importantly, we showed that, contrary to glucocorticoids that reduce the myogenic cell pool, RvD2 promotes the generation of differentiated myoblasts and their fusion and improves their overall myogenesis capacity. This pro-myogenic effect of RvD2 is mediated by the Gpr18 receptor expressed by myogenic cells. To assess the therapeutic potential of RvD2 in vivo, we followed the guidelines established by TREAT-NMD, a network of excellence on neuromuscular diseases, for preclinical research on muscular dystrophies[31]. Particularly, (1) we compared the efficacy of RvD2 to a standard drug (prednisone) for which the effects are already known; (2) we assessed recognized key parameters (grip strength, hang test, ex vivo isometric force, and validated immunohistological measurements) using standard operating procedures; and (3) we used two different mouse models of DMD, the well-characterized dystrophin-null mice (mdx mice) that recapitulate the genetic mutation observed in humans, and the utrophin-dystrophin double knockout mice (mdx-utrn dKO mice) that mimic more closely the severity of the human disease. Our findings showed that RvD2 reduced the phenotype severity and enhanced muscle function compared to untreated dystrophic mice in the short and long term. More importantly, RvD2 had a similar anti-inflammatory and anti-fibrotic capacity compared to prednisone, but its capacity to target myogenic cells and promote myogenesis led to a superior therapeutic impact on muscle function compared to glucocorticoids. Therefore, RvD2 is a promising therapeutic avenue to improve the efficacy of the gold-standard treatment for DMD.

## Results

**RvD2 induces a switch in macrophage phenotypes and promotes their release of pro-myogenic factors.** Considering the role of RvD2 in the regulation of inflammation, we first studied the capacity of RvD2 to target macrophages and promote their anti-inflammatory and pro-regenerative phenotype. Cells were collected from the bone marrow of mdx mice, and monocytes were purified by magnetic-activated cell sorting (MACS). These cells were differentiated into macrophages in vitro (Macrophage Colony-Stimulating factor; M-CSF), and polarized into

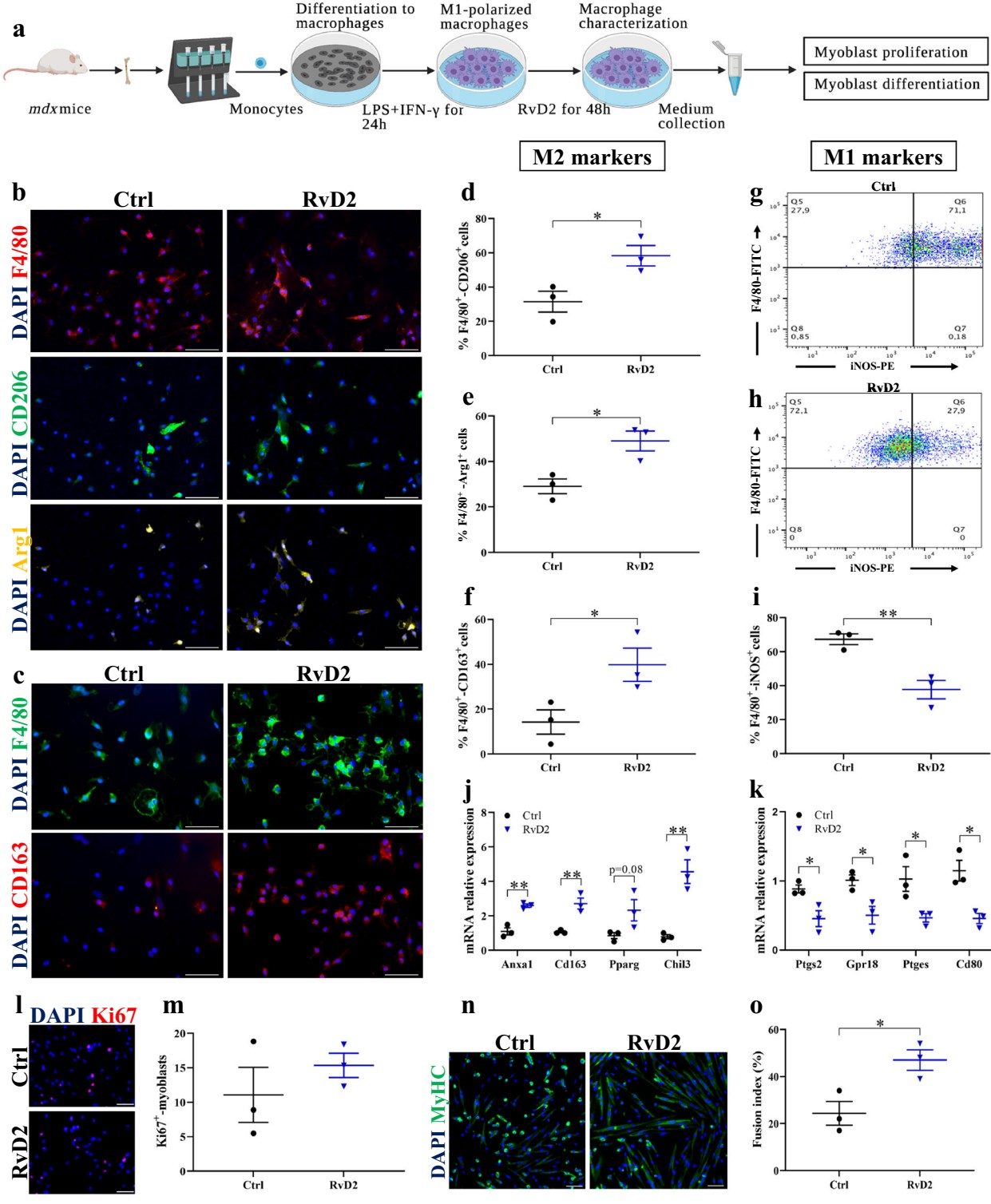

pro-inflammatory M1 macrophages (stimulation with interferon-γ and lipopolysaccharide; IFN-γ + LPS) (Fig. 1a). Macrophages were treated with RvD2 (200 nM) or vehicle for 48 h. Immunofluorescence staining showed that RvD2 treatment increased by ~ 2-fold the proportion of macrophages (F4/80[+]) expressing the anti-inflammatory macrophage markers CD206, Arginase-1 (Arg1), and CD163 (Fig. 1b–f)[32,33]. Analysis of macrophage phenotype by flow cytometry revealed that RvD2 treatment strongly decreased the proportion of macrophages expressing iNOS (inducible nitric oxide synthase), a well-known M1

macrophage marker (Fig. 1g–i)[34]. Considering that macrophages are on a continuum of polarization states, we further assessed by qPCR the expression of several key genes that were identified by transcriptome analysis to be specifically expressed in the in vivo and in vitro signatures of M1 or M2 macrophages[35,36]. Genes known to be highly enriched in the signature of M2 macrophages such as *Cd163*, *Pparg*, *Chil3*, and *Anxa1* were upregulated by RvD2 treatment (Fig. 1j). Conversely, genes that were shown to be overexpressed in the signature of M1 macrophages such as *Cd80*, *Gpr18*, *Ptgs2* (gene encoding for COX-2), and *Ptges*

**Fig. 1 Resolvin-D2 promotes the switch in macrophage phenotype and their release of pro-myogenic factors. a** Graphical overview of the myoblast:macrophage-conditioned medium co-culture experiment (image created with BioRender.com). Monocytes purified from bone marrow of *mdx* mice are differentiated into macrophages (M-CSF), polarized into M1 macrophages (interferon-gamma + lipopolysaccharide; IFN-γ and LPS**)** for 24 h and treated with Resolvin-D2 (RvD2; 200 nM) for 48 h. **b** Representative images of immunofluorescence performed on macrophages in vitro for F4/80 (pan-macrophage marker; red), and the anti-inflammatory markers CD206 (green), and Arginase-1 (Arg1; yellow), and **c** for F4/80 (green) and CD163 (red), and DAPI (blue). Scale bars = 50 μm. **d–f** Percentage of total macrophages (F4/80$^+$ cells) co-expressing the anti-inflammatory macrophages markers CD206 (**d**) ($p = 0.0339$), or Arg1 (**e**) ($p = 0.0217$), or CD163 (**f**) ($p = 0.0496$) following 48 h treatment with RvD2 (blue triangles) or control (black circles). **g, h** Representative FACS plots showing the expression of the pan-macrophage marker F4/80 (FITC, y-axis) and the pro-inflammatory marker iNOS (PE, x-axis) on M1-polarized macrophages treated with RvD2 or vehicle. **i** Quantification of the proportion of F4/80$^+$ iNOS$^+$ macrophages ($p = 0.0075$). **j, k** Gene expression of **j** the anti-inflammatory markers Anxa1($p = 0.0028$), Cd163 ($p = 0.0069$), Pparg ($p = 0.0848$), and Chil3 ($p = 0.0056$) and **k** the pro-inflammatory markers Ptgs2 ($p = 0.0290$), Gpr18 ($p = 0.0275$), Ptges ($p = 0.0404$), and Cd80 ($p = 0.0142$) on M1-polarized macrophages treated with or without RvD2 for 48 h. **l** Representative images and **m** quantification of Ki67 immunostaining (red) performed on primary myoblasts incubated with conditioned medium from macrophages treated with RvD2 or vehicle. Scale bars = 50 μm. **n** Representative images of myoblasts differentiated into myotubes for 4 days with the macrophage-conditioned medium and stained for myosin heavy chain (MyHC; green) and DAPI (blue). Scale bars = 50 μm. **o** Quantification of the fusion index (proportion of nuclei into multinucleated myotubes/total nuclei) ($p = 0.0273$). Data are presented as mean ± SEM, $n = 3$ biologically independent samples performed in technical duplicates and analyzed with the two-tailed unpaired Student's t-test. All data were analyzed with a 95% confidence interval. *$p < 0.05$, **$p < 0.01$.

(prostaglandin-E synthase) were significantly reduced by RvD2 treatment (Fig. 1k). Notably, the gene signature of RvD2-treated macrophages was very similar to the one induced by prednisone, a potent anti-inflammatory glucocorticoid that is widely used for the treatment of DMD (Supplementary Fig. 1).

To assess the ability of RvD2 to stimulate myogenesis through macrophages, we cultured *mdx* primary myoblasts with the macrophage-conditioned medium. The RvD2-treated macrophage-conditioned medium did not affect cell proliferation, but it increased the fusion index compared to the untreated macrophage medium (Fig. 1l–o). Similar results were obtained with non-polarized M0 macrophages, in which RvD2 treatment promoted the switch in macrophage phenotype towards the M2 phenotype and stimulated their expression of pro-myogenic factors that enhanced myoblast fusion and myotube growth (Supplementary Fig. 2). To ensure that the myogenic effect is not mediated by the remaining RvD2 in the macrophage-conditioned media, we analyzed RvD2 levels in the culture medium after 48 h in culture. We observed that RvD2 had been mostly degraded after 48 h (10% of its initial concentration; Supplementary Fig. 3a), consistent with the relatively short half-life of Resolvins of a few hours[29,37]. Further, we performed an additional experiment in which the RvD2-treated medium was incubated without macrophages for 48 h before being added to myoblasts. These results showed that the low levels of remaining RvD2 did not increase myogenesis compared to the control (Supplementary Fig. 3b).

**RvD2 directly targets myogenic cells to stimulate their myogenesis capacity.** To determine the direct effect of RvD2 on myogenic cells, we isolated dystrophin-deficient MuSC from *mdx* mice that we cultured in vitro with RvD2, prednisone, or vehicle. Confluence assay using an automated live imaging analysis system (IncuCyte, Essen Bioscience) revealed that RvD2 did not have a substantial impact on cell proliferation, in contrast to prednisone that reduced cell proliferation (Fig. 2a). To assess the impact of RvD2 on cell differentiation, myoblasts were cultured overnight (16 h) in a low serum medium with RvD2, prednisone, or vehicle to quantify the proportion of non-differentiated cells (Pax7$^+$ cells) compared to differentiated myoblasts (Myog$^+$ cells) by immunofluorescence. RvD2 supplementation significantly promoted myoblast differentiation, as evidenced by a 2-fold increase in myogenin-expressing differentiated myoblasts and a concomitant decrease in the proportion of Pax7$^+$ cells compared to prednisone or control (Fig. 2b–d). Furthermore, we used isolated single myofibers as another in vitro model to study the myogenic progression of MuSC in their physiological niche. Single myofibers

isolated from the *extensor digitorium longus* (EDL) muscle of *mdx* mice were cultured for 72 h with RvD2, prednisone, or vehicle. We observed that RvD2 did not significantly affect the absolute number of MuSC/proliferating myoblasts (Pax7$^+$) but induced a 1.7-fold increase in the number of differentiated myoblasts (Myog$^+$) compared to the control (Fig. 2e–g). These changes resulted in a significant increase in the proportion of Myog$^+$ cells, indicating that RvD2 pushes myoblasts towards differentiation (Fig. 2h, i). On the opposite, prednisone reduced both the number of Pax7$^+$ and Myog$^+$ cells in treated fibers (Fig. 2e–g). To assess the impact of RvD2 on myoblast fusion and myotube growth, we differentiated primary *mdx* myoblasts into myotubes in a low-serum medium supplemented with RvD2, prednisone, or vehicle for 4 days. Western blot and immunofluorescence analysis revealed that RvD2 treatment increased the fusion index and the expression of Myosin Heavy Chain (MyHC), a marker of myotube formation, compared to control and prednisone treatment (Fig. 2j–m). The addition of RvD2 on myotubes that were already formed for 2 days in the differentiation medium also induced an increase in myotube size, albeit at a lower level compared to when RvD2 was added at the start of the differentiation process (Supplementary Fig. 4).

Specialized pro-resolving mediators were shown to bind to specific G protein-coupled receptors[38]. Particularly, Gpr18 has been identified on inflammatory cells as the main receptor that transduces intracellular RvD2 signals[39]. Immunostaining analysis of Gpr18 expression on inflammatory cells in skeletal muscle confirmed that this receptor is expressed in a large subpopulation of macrophages, which likely correspond to the pro-inflammatory macrophages, considering that Gpr18 has been shown to be strongly enriched in M1 macrophages (Supplementary Fig. 5a)[35,36]. Gpr18 expression was not observed in Tregs (CD25$^+$ cells[40]), consistent with its low expression in muscle Tregs compared to spleen Tregs (Supplementary. Fig. 5b, d)[40]. We further investigated the expression of Gpr18 on two muscle-resident cell types, the myogenic cells and the fibroadipogenic progenitors (FAPs). Western blot showed that Gpr18 is expressed in proliferating myoblasts, and that its expression increases during myoblast differentiation (Fig. 3a, b). Immunofluorescence confirmed that Myog$^+$ cells express Gpr18 (Supplementary Fig. 5c, d). On the other hand, the expression of Gpr18 was very low in FAPs compared to proliferating myoblasts (8-fold lower); and RvD2 treatment had no effect on FAPs proliferation (Supplementary Fig. 5e–g). Considering these facts, we focused on myogenic cells for subsequent experiments. We performed a knockdown of this receptor in dystrophin-deficient myoblasts to determine if it affects the myogenic activity of RvD2 (Fig. 3c).

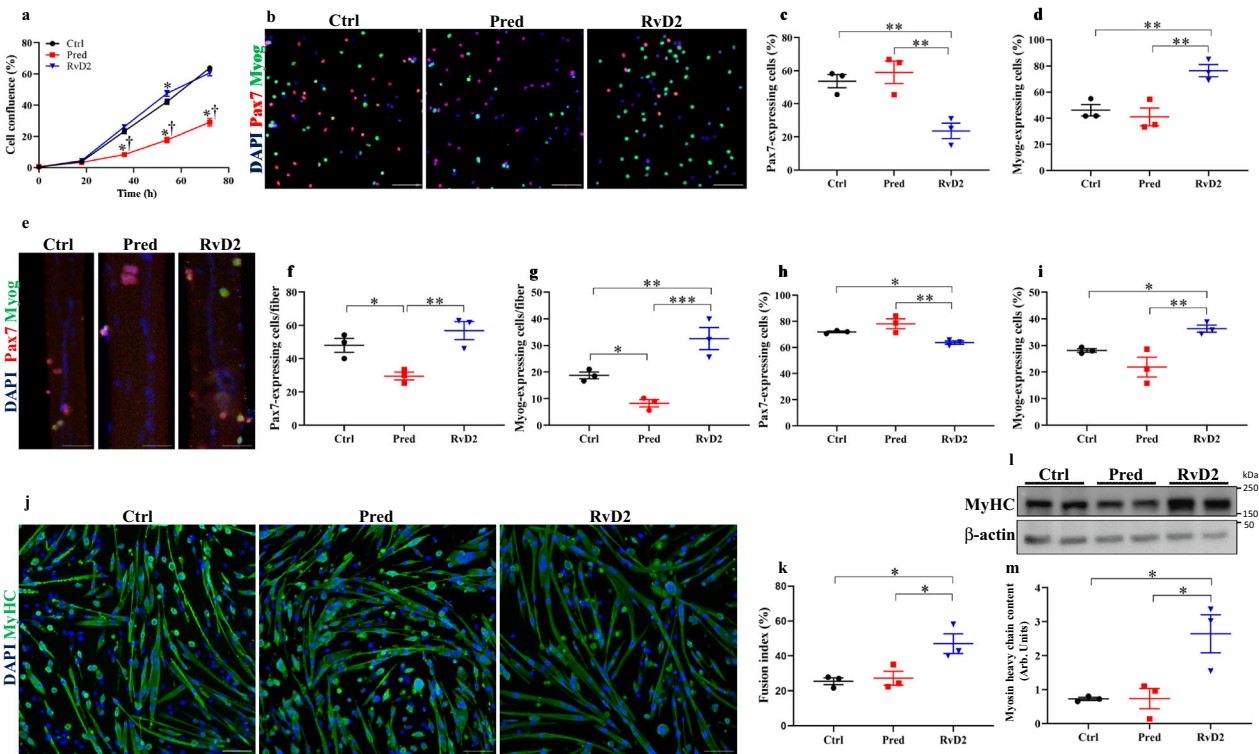

**Fig. 2 Resolvin-D2 directly targets myogenic cells and stimulates their myogenesis capacity. a** Primary myoblasts isolated from *mdx* mice were plated in vitro in an automated live imaging analysis system (IncuCyte). Cells were cultured in proliferation media supplemented with Resolvin-D2 (RvD2, 200 nM, blue triangles), prednisone (pred, 10 μM, red squares), or vehicle (Ctrl, black circles), and cell confluence was assessed for 3 days (36 h: Ctrl vs. Pred, $p < 0.0001$; Ctrl vs. RvD2, $p = 0.1352$; Pred vs. RvD2, $p < 0.0001$; 54 h: Ctrl vs. Pred, $p < 0.0001$; Ctrl vs. RvD2, $p = 0.0084$; Pred vs. RvD2, $p < 0.0001$; 72 h: Ctrl vs. Pred, $p < 0.0001$; Ctrl vs. RvD2, $p = 0.1089$; Pred vs. RvD2, $p < 0.0001$). **b–d** Dystrophin-deficient myoblasts were cultured in low serum medium for 16 h supplemented with RvD2, prednisone, or vehicle. **b** Representative images of immunofluorescence for Pax7 (red) and Myog (green). Scale bars = 100 μm. **c, d** Quantification of the proportion of Pax7-expressing cells (MuSC/proliferative myoblasts) (Ctrl vs. Pred, $p = 0.4995$; Ctrl vs. RvD2, $p = 0.0070$; Pred vs. RvD2, $p = 0.0032$) and Myog-expressing cells (differentiated myoblasts) (Ctrl vs. Pred, $p = 0.5243$; Ctrl vs. RvD2, $p = 0.0074$; Pred vs. RvD2, $p = 0.0035$). **e–i** Single myofibers were isolated from the EDL muscle of *mdx* mice and cultured for 72 h with RvD2 (200 nM), prednisone (10 μM), or vehicle ($n = 30$ fibers/biological sample). **e** Representative images of myofibers immunostained for Pax7 (red), Myog (green), and DAPI (blue). Scale bars = 50 μm. **f, g** Quantification of the total number of Pax7+ (Ctrl vs. Pred, $p = 0.0204$; Ctrl vs. RvD2, $p = 0.1893$; Pred vs. RvD2, $p = 0.0037$) cells and Myog+ (Ctrl vs. Pred, $p = 0.03$; Ctrl vs. RvD2, $p = 0.0098$; Pred vs. RvD2, $p = 0.0006$) cells per fiber. **h, i** Proportion of pax7-expressing and myogenin-expressing cells (Ctrl vs. Pred, $p = 0.1040$; Ctrl vs. RvD2, $p = 0.0466$; Pred vs. RvD2, $p = 0.0045$) relative to the total number of myogenic cells (Pax7+ and Myog+). **j, k** Dystrophin-deficient myoblasts were differentiated for 4 days in low serum medium supplemented with RvD2, prednisone, or vehicle. **j** Representative images of myotubes immunostained for MyHC (green) and DAPI (blue). Scale bars = 75 μm. **k** Quantification of the fusion index (proportion of nuclei into multinucleated myotubes/total nuclei) (Ctrl vs. Pred, $p = 0.7713$; Ctrl vs. RvD2, $p = 0.0101$; Pred vs. RvD2, $p = 0.0146$). **l** Representative images and **m** quantification of myosin heavy chain expression by Western blot in myotubes (relative to β-actin as loading control) (Ctrl vs. Pred, $p = 0.9899$; Ctrl vs. RvD2, $p = 0.0102$; Pred vs. RvD2, $p = 0.0103$). The samples derive from the same experiment and the gels/blots were processed in parallel. **a, c, f–i, k, m** RvD2 = blue triangles, pred = red squares or Ctrl = black circles. Data are presented as mean ± SEM, $n = 3$ biologically independent samples performed in technical duplicates and analyzed with one-way ANOVA uncorrected Fisher's LSD test. All data were analyzed with a 95% confidence interval. *$p < 0.05$ compared with the vehicle and † $p < 0.05$ compared with prednisone for panel (**a**). *$p < 0.05$, **$p < 0.01$, ***$p < 0.001$ for **c, d, f–i, k**, and **m**.

Knockdown of Gpr18 blocked the ability of RvD2 to increase myotube fusion and growth (Fig. 3d–f). To determine the downstream signaling of Gpr18, we treated myoblasts with RvD2 for 0, 5, 15, 30, 60, or 120 min (Fig. 3g–j). This time course experiment revealed that the addition of RvD2 into the media of *mdx* myoblasts induced rapid activation of the Akt-1 pathway (phosphorylation on Ser[473]) that peaked after 60 min (Fig. 3g, h). This transient rise in the phosphorylation of Akt-1 is not observed in primary myoblasts isolated from Gpr18-knockout mice (Fig. 3i, j).

**RvD2 dampens inflammation in dystrophic *mdx* mice.** To determine the ability of RvD2 to dampen the chronic and excessive inflammatory process in vivo in dystrophic muscles, we used *mdx* mice, a well-characterized model of DMD[41–43]. We first investigated the expression of Resolvins and other bioactive lipids in the skeletal muscles of these mice compared to healthy wild-type (WT) mice. Results from mass spectrometry experiments (LC-MS/MS) indicated that the different isoforms of Resolvins-D (RvD1, RvD2, RvD3, 17(R, S)-RvD4, RvD5) are expressed at similar levels between WT and *mdx* mice (Fig. 4a and Supplementary Fig. 6). However, the expression of the pro-inflammatory eicosanoids, such as prostaglandins ($PGE_2$ and $PGF_{2\alpha}$), leukotriene-B4 (LTB4), and thromboxane-B2 (TXB2) is increased from roughly 3-fold in *mdx* mice (Fig. 4b and Supplementary Fig. 6). These changes led to an increase in the prostaglandins/Resolvins ratio (2.5-fold higher in *mdx* compared to WT), indicating an unbalance in favor of pro-inflammatory signals in the muscles of *mdx* mice (Fig. 4c).

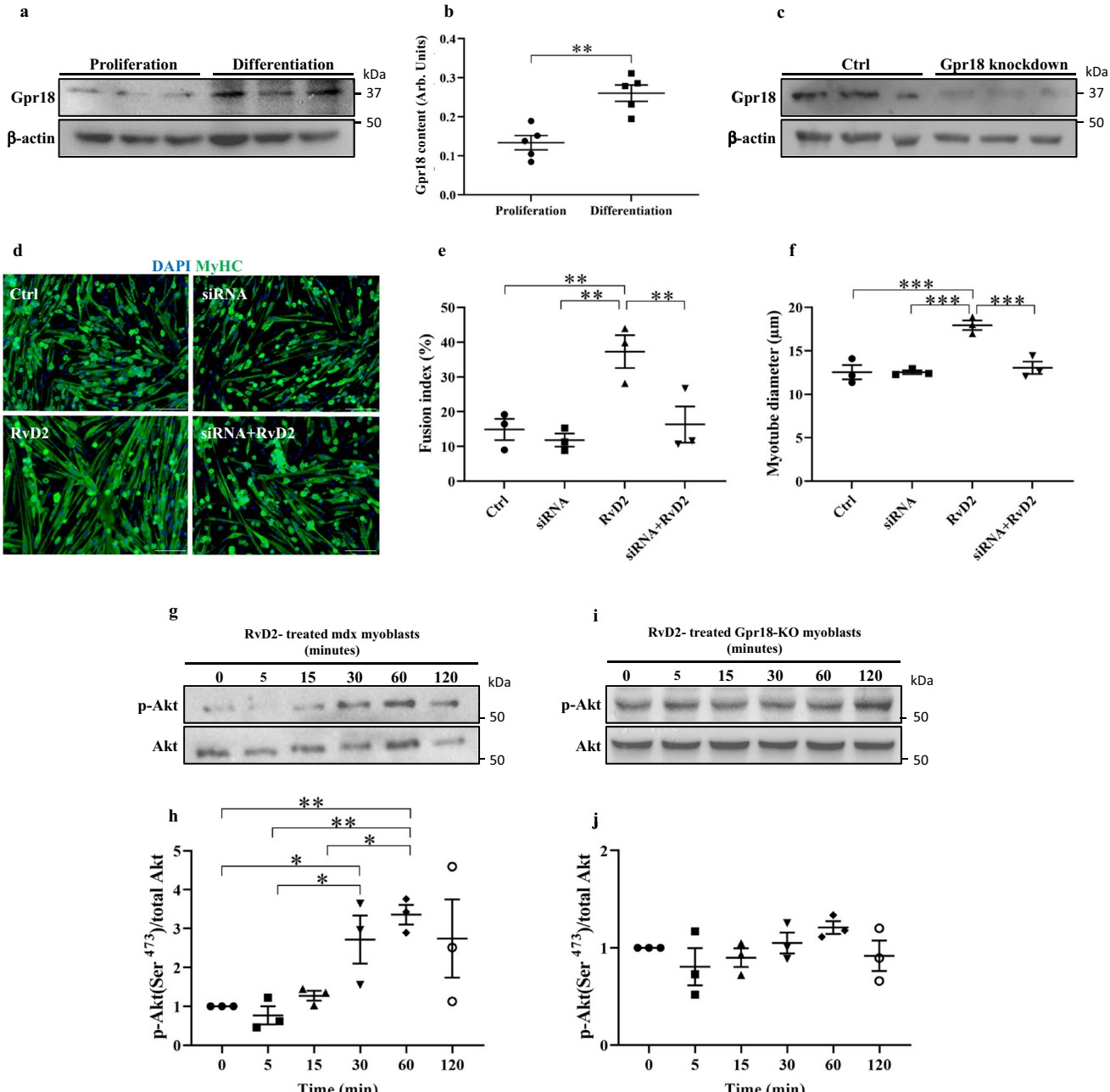

**Fig. 3 The effect of Resolvin-D2 on the myogenesis capacity of myogenic cells is mediated by Gpr18. a** Representative Western blot and **b** quantification of Gpr18 expression in proliferating myoblasts and differentiated myoblasts (1 day in differentiating medium) isolated from *mdx* mice (data are presented as mean ± SEM, *n* = 5 biologically independent samples, two-tailed unpaired Student's *t*-test; *p* = 0.0018). **c** siRNA knockdown of Gpr18 on primary myoblasts isolated from *mdx* mice. **d**–**f** Dystrophin-deficient myoblasts were treated with siRNA-scrambled (Ctrl) or siRNA-Gpr18 (siRNA) and cultured in differentiating media containing Resolvin-D2 (RvD2, 200 nM) or not for 4 days. **d** Representative images of myotubes stained for MyHC. Scale bars = 75 μm. Quantification of (**e**) the fusion index (siRNA vs. Ctrl, *p* = 0.5967; RvD2 vs. Ctrl, *p* = 0.0038; siRNA+RvD2 vs. Ctrl, *p* = 0.8042; RvD2 vs. siRNA, *p* = 0.0018; siRNA+RvD2 vs. siRNA, *p* = 0.4428; siRNA+RvD2 vs. RvD2, *p* = 0.0055), and **f** the myotube diameter (siRNA vs. Ctrl, *p* = 0.9937; RvD2 vs. Ctrl, *p* = 0.0003; siRNA+RvD2 vs. Ctrl, *p* = 0.5762; RvD2 vs. siRNA, *p* = 0.0003; siRNA+RvD2 vs. siRNA, *p* = 0.5710; siRNA+RvD2 vs. RvD2, *p* = 0.0005). **g**–**j** Representative Western blot and quantification of the p-Akt (phosphorylation on Ser473)/total Akt-1 ratio in primary myoblasts isolated from **g**, **h** *mdx* mice (0 vs. 30 min, *p* = 0.0329; 0 vs. 60 min, *p* = 0.0062; 5 vs. 30 min, *p* = 0.0181; 5 vs. 60 min, *p* = 0.0034; 15 vs. 60 min, *p* = 0.0128) or **i**, **j** Gpr18-knockout mice cultured for 0, 5, 15, 30, 60, or 120 min with RvD2. For Western blot experiments, the samples derive from the same experiment and the gels/blots were processed in parallel. Data are presented as mean ± SEM, *n* = 3 biologically independent samples performed in technical duplicates and analyzed with one-way ANOVA uncorrected Fisher's LSD test (**e**, **f**, **h**, and **j**). All data were analyzed with a 95% confidence interval. \**p* < 0.05, \*\**p* < 0.01, \*\*\**p* < 0.001.

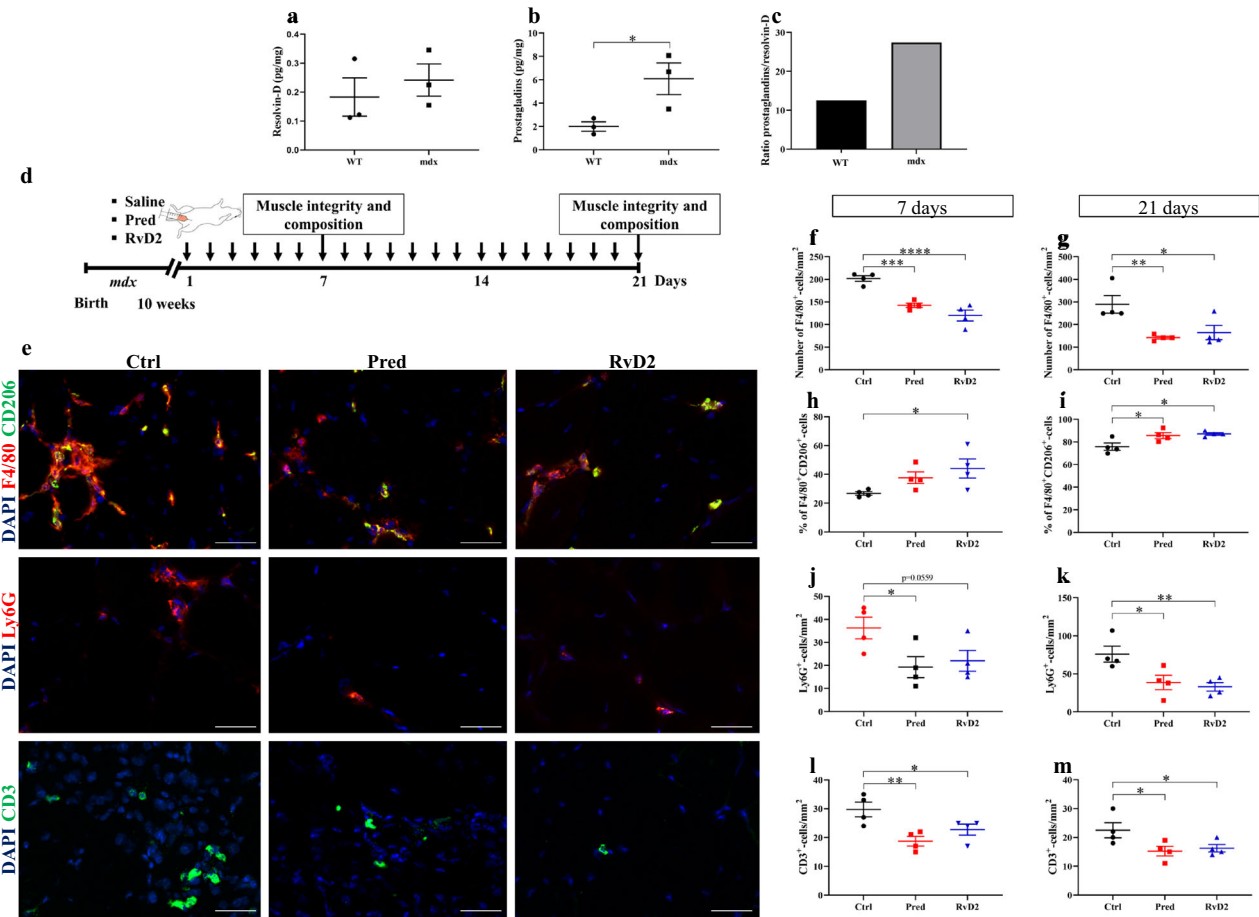

**Fig. 4 Resolvin-D2 regulates inflammation in dystrophic *mdx* mice. a, b** Quantification by mass spectrometry (LC-MS/MS) of Resolvin-D isoforms (RvD1, RvD2, RvD3, 17(R, S)-RvD4, RvD5), and ptgs1/2-derived prostaglandins (PGE$_2$, PGF$_{2\alpha}$) in the gastrocnemius muscle of wildtype (WT) and *mdx* mice (**b**) $p = 0.0444$. **c** Ratio of prostaglandins/Resolvin-D in the gastrocnemius muscle of wildtype and *mdx* mice. **d** Timeline of daily intraperitoneal (*ip*) injection of Resolvin-D2 (RvD2, 5 µg/kg/day), prednisone (pred, 2 mg/kg/day), or vehicle (ctrl) in *mdx* mice. **e** Immunofluorescence on tibialis anterior (TA) muscle sections of *mdx* mice for F4/80 (total macrophage marker; red) and CD206 (anti-inflammatory macrophage marker; green), Ly6G (neutrophil marker; red), and CD3 (lymphocyte marker; green), and DAPI (blue). Scale bars = 50 µm. **f, g** Quantification of total macrophages in TA muscles following 7 days **f** (Ctrl vs. Pred, $p = 0.0007$; Ctrl vs. RvD2, $p < 0.0001$; Pred vs. RvD2 = 0.0835), and 21 days **g** (Ctrl vs. Pred, $p = 0.0060$; Ctrl vs. RvD2, $p = 0.0141$; Pred vs. RvD2, $p = 0.6009$). **h, i** Percentage of anti-inflammatory macrophages (F4/80+ CD206+) in TA muscles sections at **h** 7 days (Ctrl vs. Pred, $p = 0.1264$; Ctrl vs. RvD2, $p = 0.0249$; Pred vs. RvD2, $p = 0.3418$) and **i** 21 days (Ctrl vs. Pred, $p = 0.0212$; Ctrl vs. RvD2, $p = 0.0104$; Pred vs. RvD2, $p = 0.6724$). **j, k** Density of neutrophils in TA muscles at **j** 7 days (Ctrl vs. Pred, $p = 0.0279$; Ctrl vs. RvD2, $p = 0.0559$; Pred vs. RvD2, $p = 0.6820$) and **k** 21 days (Ctrl vs. Pred, $p = 0.0151$; Ctrl vs. RvD2, $p = 0.0072$; Pred vs. RvD2, $p = 0.6548$). **l, m** Density of CD3+ T cells in TA muscles at **l** 7 days (Ctrl vs. Pred, $p = 0.0047$; Ctrl vs. RvD2, $p = 0.0415$; Pred vs. RvD2, $p = 0.2076$) and **m** 21 days (Ctrl vs. Pred, $p = 0.0272$; Ctrl vs. RvD2, $p = 0.0494$; Pred vs. RvD2, $p = 0.7249$). **f–m** RvD2 = blue triangles, pred = red squares, and Ctrl = black circles. Data are presented as mean ± SEM, $n = 4$ mice per group per time-point (except $n = 3$ for **a–c**). Experiments were performed in technical duplicates and analyzed using two-tailed unpaired Student's *t*-test for panels (**a, b**) and one-way ANOVA uncorrected Fisher's LSD test for (**f–m**). All data were analyzed with a 95% confidence interval. *$p < 0.05$; **$p < 0.01$; ***$p < 0.001$; ****$p < 0.0001$.

To determine the therapeutic potential of RvD2 for the treatment of DMD, the effect of a daily systemic delivery of RvD2 was compared to a control and the gold-standard treatment for DMD, i.e., glucocorticoids (prednisone) (Fig. 4d). RvD2 was delivered at a dose of 5 µg/kg/day. Similar dosages were used in previous studies and were shown to induce a physiological increase in the level of this Resolvin in the plasma or skeletal muscles a few hours after the injection[29,37]. Prednisone, one of the most widely used glucocorticoids for the treatment of DMD, was administered at a dose of 2 mg/kg/day, which is similar to what has been used in previous studies on *mdx* mice and is representative of the dosage used in humans[16,44,45]. Our results showed that RvD2 closely mimicked the anti-inflammatory effect observed with prednisone, as they similarly decreased the accumulation of F4/80+ macrophages by 40–50% after 7 and

21 days of daily systemic injection compared to the control group (Fig. 4e–g). Moreover, the percentage of anti-inflammatory macrophages (F4/80+/CD206+) was higher in the RvD2-treated and the prednisone-treated mice as compared to the control group at 21 days (Fig. 4e, h, i). Consistently, densities of Ly6G+ neutrophils and CD3+ T cells were reduced to a similar level in muscle cross sections of *mdx* mice treated with either RvD2 or prednisone for 7 and 21 days compared to the control (Fig. 4e, j–m). Of note, the density of Tregs (CD3+Foxp3+ cells) was not affected by RvD2 or prednisone treatment (Supplementary Fig. 7). No additive effect was observed with the combination of both prednisone and RvD2 on the inflammatory cell accumulation (Supplementary Fig. 8). Moreover, to determine whether RvD2 acts through local or systemic actions, we performed a single intramuscular (i.m.) administration of RvD2 within the TA

muscle of *mdx* mice. Data showed that, similarly to systemic injection, local administration of RvD2 decreases the total number of F4/80$^+$ macrophages and increases the percentage of the anti-inflammatory F4/80$^+$/CD206$^+$ macrophages 7 days post injection compared to saline-injected muscles (Supplementary Fig. 9).

**RvD2 promotes myogenesis in *mdx* mice to a higher level than glucocorticoids.** To determine the potential of RvD2 to stimulate skeletal muscle regeneration in DMD, we first quantified the density of MuSC (Pax7$^+$ cells) and differentiated myoblasts (Myog$^+$ cells) in TA muscle of *mdx* mice treated with RvD2, prednisone, or saline (Fig. 5a–f, and Supplementary Fig. 10). We observed that while prednisone administration did not affect the pool of myogenic cells in dystrophic muscles, the daily administration of RvD2 enhanced by roughly 1.7-fold the total number of myogenic cells at day 21 (Supplementary Fig. 10). Further analysis revealed that this increase in the myogenic cell number is not caused by changes in the number of Pax7$^+$ MuSC but rather by a 2–3-fold increase in the number of Myog$^+$ differentiated myoblasts (Fig. 5b–f). This rise in Myog$^+$ myoblasts was observed at 7 and 21 days, indicating that the effect of RvD2 on myogenic cells is rapid and sustained. Consistent with the rise in myogenic cell number, an increase in the number of myonuclei per fiber was observed in RvD2-treated mice at 21 days, indicating that these cells have fused and donated their nuclei to regenerating fibers (Fig. 5b, g, h). Accordingly, the proportion of regenerating myofibers expressing the embryonic isoform of MyHC (MyHC-emb) is increased in RvD2-treated mice compared to control and prednisone-treated mice at days 7 and 21 (Fig. 5b, i, j). In contrast, analysis of IgG-positive fibers indicated that RvD2 did not significantly reduce myofiber necrosis compared to control (Supplementary Fig. 11). The enhanced myogenesis status in RvD2-treated muscle was associated with larger muscle fiber diameter after 7 and 21 days compared to control and prednisone-treated mice (Fig. 5b, k, l). Similar to systemic delivery, a local administration of RvD2 increased the number of Myog$^+$ differentiated myoblasts and myofiber size 7 day after a single i.m. injection in the TA muscle of *mdx* mice compared to saline-injected muscle (Supplementary Fig. 9d, e). Lastly, we assessed the impact of the different treatments on fibrosis using the diaphragm muscle (muscle most severely affected by fibrosis deposition in *mdx* mice). Our results showed that RvD2 decreases fibrosis compared to control, but to a similar extent than prednisone (Fig. 5b, m, n).

**RvD2 improves muscle function in *mdx* mice to a higher level than glucocorticoids.** To determine if the improvements in muscle phenotype translate into enhanced muscle function, standardized functional measurements were performed in *mdx* mice using different experimental setups. In the first set of experiments, daily i.p. injections of RvD2, prednisone, or saline were given to *mdx* mice for 7 or 21 days (Fig. 6a). Assessment of global physical function using the hang test (time mice can hold on an inverted grid) showed that RvD2 significantly increased global physical function at day 7 and 21 compared to control and prednisone-treated *mdx* mice (Fig. 6b, e). Thereafter, at the day of sacrifice, the EDL muscle of *mdx* mice was isolated to assess the contractile properties ex vivo. We observed an increase in the force-frequency curve for both absolute force (mN) and specific force (N/cm$^2$) in the RvD2-treated group compared to both prednisone and saline-treated *mdx* mice (Fig. 6c, d, f, g). This improvement in muscle force was observed as early as day 7 and was maintained at day 21.

In a second experimental setup, we investigated the therapeutic potential of intermittent administration (once a week) instead of daily injection of RvD2 (Fig. 6h). This strategy has been tested previously in an attempt to maximize the benefits and minimize the side effects of glucocorticoids[44]. Analysis of isometric contractile properties ex vivo revealed that similar to what was observed in the first experimental setup, RvD2 increased the absolute and specific muscle force to a higher level than the control and prednisone treatments (Fig. 6i, j). These data indicate that the efficacy of weekly administration of RvD2 on muscle force is similar to the daily injection (Fig. 6f, g, i, j).

To validate the molecular mechanism of RvD2 in vivo, the *mdx* mice were injected with a Gpr18 antagonist, O-1918, prior to the administration of RvD2 (Fig. 6k)[46–51]. Administration of O-1918 did not have an effect on vehicle-injected *mdx* mice; however, it completely ablated the therapeutic effect of RvD2 on the muscle force of *mdx* mice (Fig. 6l, m).

Next, we explored whether the therapeutic potential of RvD2 is sustained in the long term. *Mdx* mice were treated weekly with RvD2, prednisone, or vehicle for 2 months (Fig. 7a). Analysis of the myogenic cell pool indicated that long-term treatment of RvD2 did not lead to an exhaustion of the MuSC pool, as the number of Pax7$^+$ is similar to control mice (Fig. 7b, c). Notably, the number of Pax7$^+$ cells is increased in RvD2-treated *mdx* mice compared to prednisone-treated mice. Similar to what was observed at 1 and 3 weeks of treatment, the number of Myog$^+$ cells is increased by ~ 3-fold in RvD2-treated mice compared to prednisone-treated and control mice after 2 months of treatment (Fig. 7b, d). These results were accompanied by an increase in the number of myonuclei per fiber in RvD2-treated mice compared to prednisone-treated and control mice (Fig. 7b, e). As observed at 1 and 3 weeks post-treatment, the proportion of fibrotic tissue in the diaphragm muscle was decreased in RvD2 and prednisone-treated mice compared to control after 2 months of treatment (Fig. 7b, f). Analysis of muscle force in vivo after 2 months of treatment indicated a higher muscle strength in RvD2-treated mice compared to prednisone-treated and control mice (Fig. 7g). Assessment of ex vivo contractile properties confirmed the increase in muscle force of RvD2-treated mice compared to prednisone-treated and control mice (Fig. 7h).

**The potent therapeutic impact of RvD2 is maintained in the severely affected *mdx-utrn* dKO mouse model.** While the *mdx* mice are a well-characterized model of DMD, this model does not capture the severity of the human phenotype. The use of a second model is generally recommended for preclinical studies on DMD. Therefore, we used the *mdx-utrn* dKO mice, which are deficient in dystrophin and utrophin, as a more severe mouse model of DMD[52]. Mice were injected weekly with RvD2, prednisone, or vehicle for 3 weeks (Fig. 8a). Similar to our findings in *mdx* mice (Fig. 4d–i), a decrease in the number of macrophages and a switch toward their anti-inflammatory phenotype was observed in RvD2- and prednisone-treated mice compared to the control (Fig. 8b–d). Likewise, RvD2 administration increased the number of Myog$^+$ cells by over 2-fold compared to prednisone-treated and control mice, without affecting the number of Pax7$^+$ cells (Fig. 8b, e, f). Analysis of muscle function in vivo showed an increase in hanging time in RvD2-treated mice compared to prednisone and control-treated mice (Fig. 8g). Finally, ex vivo contractile properties confirmed the increase in muscle force of RvD2-treated mice compared to prednisone and control-treated mice (Fig. 8h).

## Discussion

Our findings demonstrate that RvD2 administration to dystrophic mice dampens inflammation, reduces fibrosis and

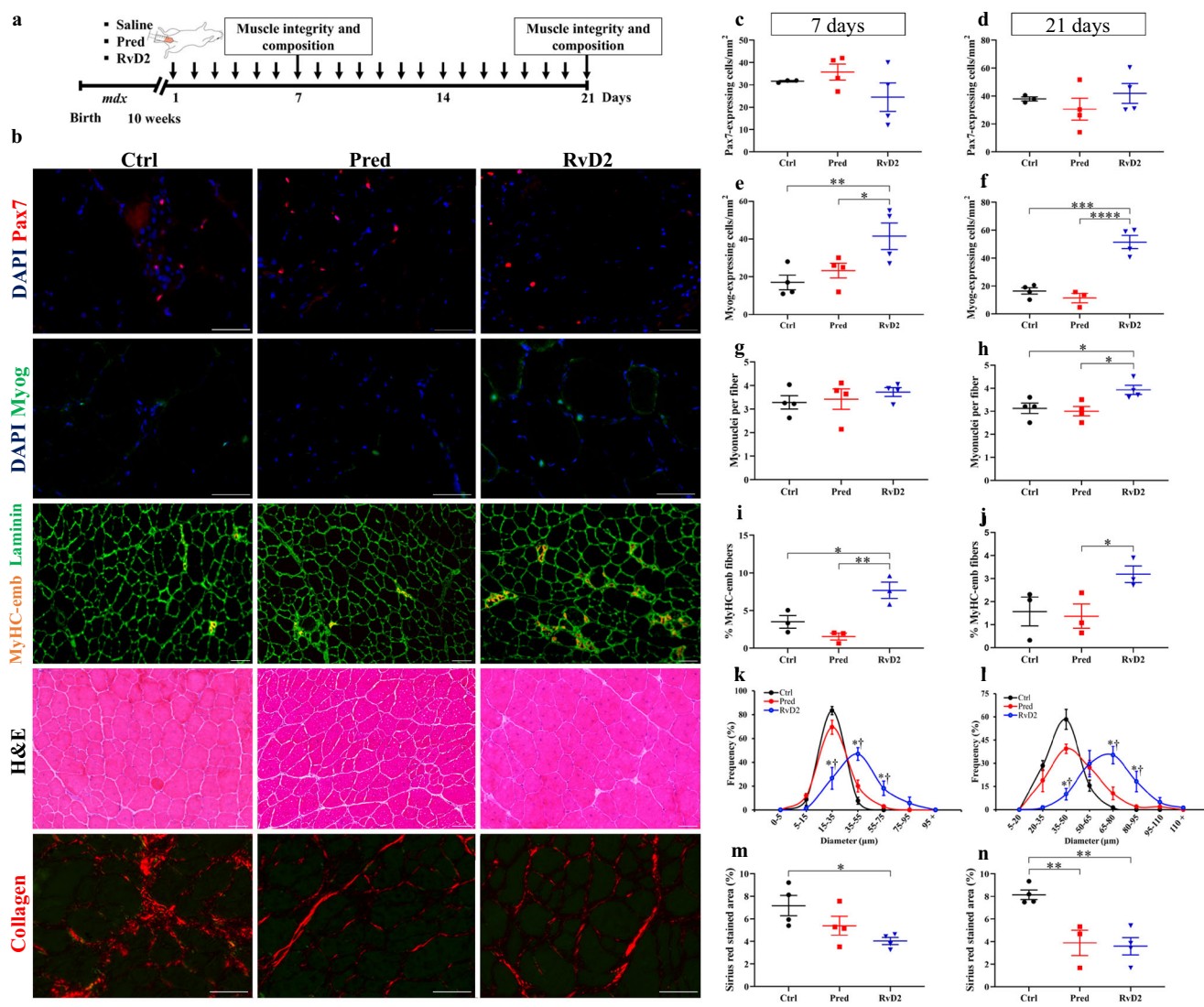

**Fig. 5 Resolvin-D2 promotes myogenesis to a higher level than glucocorticoids in dystrophic *mdx* mice. a** Timeline of daily intraperitoneal (*ip*) injection of Resolvin-D2 (RvD2, 5 ug/kg/day), prednisone (pred, 2 mg/kg/day), or vehicle (ctrl) in *mdx* mice. **b** Representative images of Pax7 (red), Myog (green), embryonic Myosin Heavy Chain (MyHC-emb; orange) + laminin (green), H&E on muscle sections of *tibialis anterior* (TA) muscle and collagen (Sirius red; red) on diaphragm sections of *mdx* mice treated with RvD2, pred, or vehicle. Scale bars = 50 μm. **c–f** Quantification of number of Pax7$^+$ (**c, d**) and Myog$^+$ cells (**e, f**) in TA muscles of *mdx* mice treated for 7 days (**e**: Ctrl vs. Pred, $p = 0.4138$; Ctrl vs. RvD2, $p = 0.0084$; Pred vs. RvD2, $p = 0.0338$) or 21 days (**f**: Ctrl vs. Pred, $p = 0.3720$; Ctrl vs. RvD2, $p = 0.0001$; Pred vs. RvD2, $p < 0.0001$). **g, h** Number of myonuclei per fiber after 7 or 21 days of treatment (21 d: Ctrl vs. Pred, $p = 0.6880$; Ctrl vs. RvD2, $p = 0.0263$; Pred vs. RvD2, $p = 0.0134$). **i, j** Proportion of myofibers expressing embryonic Myosin heavy chain (MyHC-emb) after 7 (Ctrl vs. Pred, $p = 0.1513$; Ctrl vs. RvD2, $p = 0.0124$; Pred vs. RvD2, $p = 0.0021$) or 21 days of treatment (Ctrl vs. Pred, $p = 0.7919$; Ctrl vs. RvD2, $p = 0.0681$; Pred vs. RvD2, $p = 0.0467$). **k, l** Distribution curve of minimal fiber diameter in TA muscles of *mdx* mice treated for 7 days (15–35 μm: Ctrl or Pred vs. RvD2, $p < 0.0001$. 35–55 μm: Ctrl vs. RvD2, $p < 0.0001$; Pred vs. RvD2, $p = 0.0001$. 55–75 μm: Ctrl vs. RvD2, $p = 0.0058$; Pred vs. RvD2, $p = 0.0181$) or 21 days (35–50 μm: Ctrl or Pred vs. RvD2, $p < 0.0001$. 65–80 μm: Ctrl or Pred vs. RvD2, $p < 0.0001$. 80–95 μm: Ctrl vs. RvD2, $p = 0.0035$; Pred vs. RvD2, $p = 0.0101$). **m, n** Proportion of diaphragm muscle section stained with Sirius red (collagen) after 7 days (Ctrl vs. Pred, $p = 0.1176$; Ctrl vs. RvD2, $p = 0.0141$; Pred vs. RvD2, $p = 0.2234$) or 21 days of treatment (Ctrl vs. Pred, $p = 0.0050$; Ctrl vs. RvD2, $p = 0.0022$; Pred vs. RvD2, $p = 0.7956$). **c–n** RvD2 = blue triangles, pred = red squares, and Ctrl = black circles. Data are presented as mean ± SEM, $n = 4$ mice per group/time-point (except $n = 3$ for **i, j, k**, and **l**; Ctrl group for **c** and **d**; Pred group for **f** and **n**). Experiments were performed in technical duplicates using one-way ANOVA uncorrected Fisher's LSD test (two-way ANOVA for **k** and **l**). All data were analyzed with a 95% confidence interval. *$p < 0.05$; **$p < 0.01$; ***$p < 0.001$; ****$p < 0.0001$ for **c–j**, **m**, and **n**. *$p < 0.05$ compared with the vehicle; †$p < 0.05$ compared with prednisone for **k** and **l**.

increases the number of myogenic cells, muscle fiber size, muscle force, and global physical function compared to non-treated mice. More importantly, these beneficial effects of RvD2 are more potent than the ones of glucocorticoids, the gold-standard treatment for DMD. Considering that RvD2 and glucocorticoids have similar anti-inflammatory and anti-fibrotic effects on dystrophic muscles, our findings suggest that the superior therapeutic effect

of RvD2 is mostly attributable to its capacity to directly target myogenic cells in order to improve myogenesis.

Accumulating evidence indicates that the progression of DMD is not only attributable to muscle degeneration but also to impairments in muscle regeneration[41]. Notably, after an acute injury, muscle regeneration is delayed in dystrophic mice compared to wild-type mice[8,9]. Studies have suggested that this

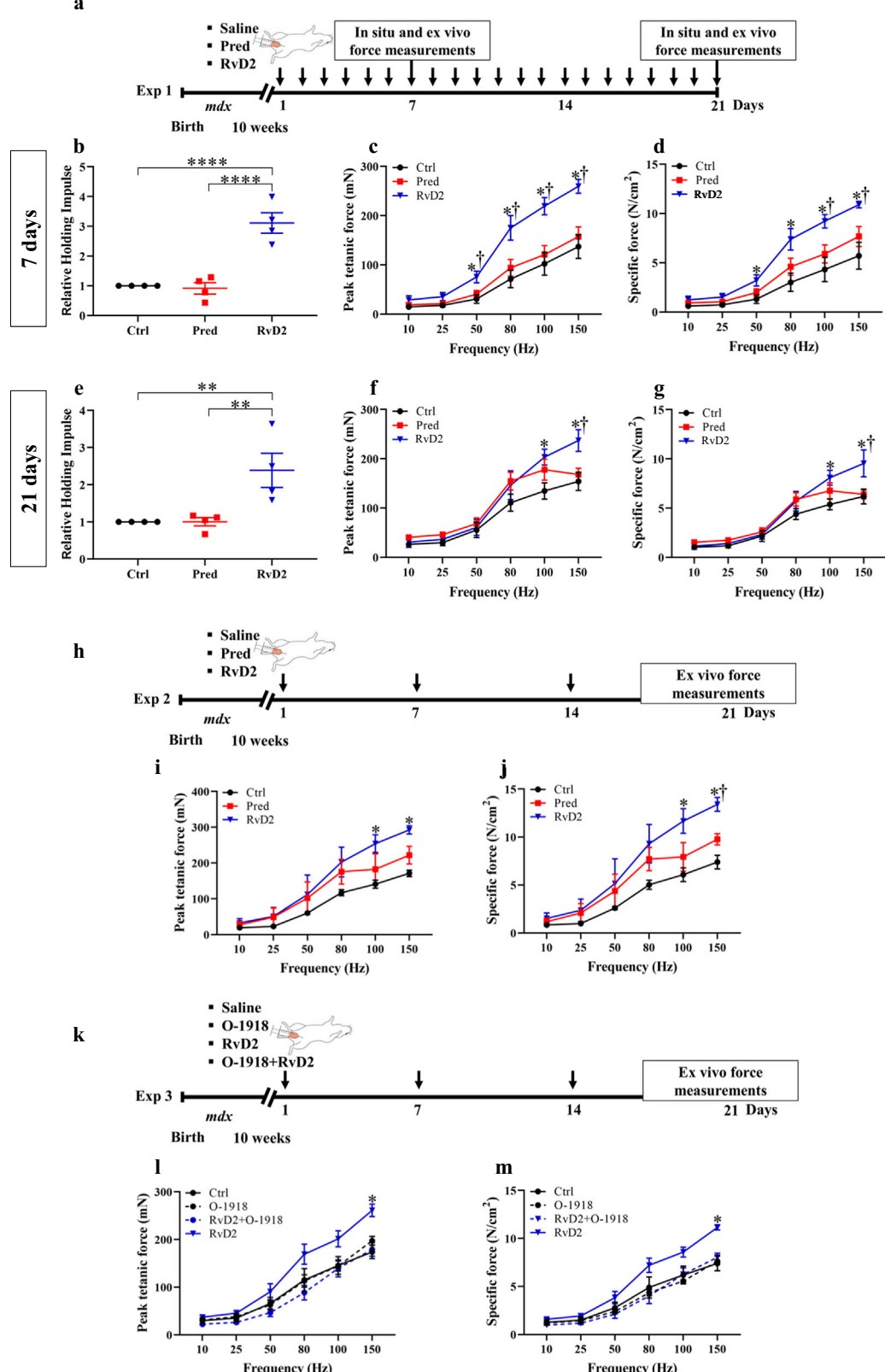

reduced regenerative capacity of skeletal muscle in DMD is caused by different factors such as the gradual exhaustion of the MuSC pool[53], as well as intrinsic and extrinsic factors impairing the myogenic capacity of the remaining cells. Extrinsic factors such as the chronic accumulation of inflammatory cells were shown to send conflicting signals to MuSC/myoblasts that impaired their myogenesis capacity[54]. Different studies have shown that specific depletion of neutrophils[55], macrophages[34,56], or lymphocytes[57,58] improved muscle phenotype in mouse models of DMD. However, while macrophages contribute to disease progression in DMD, they also play a key role in the regulation of myogenic cell proliferation and differentiation in

**Fig. 6 Resolvin-D2 improves muscle force and function to a higher level than glucocorticoids in the *mdx* mouse model. a** Timeline of daily intraperitoneal (*ip*) injection of either Resolvin-D2 (RvD2, 5 μg/kg/day), prednisone (pred, 2 mg/kg/day) or vehicle (ctrl) into *mdx* mice. **b, e** Hang test performance (time mice can hold on an inverted grid) of *mdx* mice following daily *ip* treatment for 7 days (**b**) (Ctrl vs. Pred, $p = 0.7962$; Ctrl vs. RvD2, $p < 0.0001$; Pred vs. RvD2, $p < 0.0001$), and 21 days (**e**) (Ctrl vs. Pred, $p = 0.9931$; Ctrl vs. RvD2, $p = 0.0058$; Pred vs. RvD2, $p = 0.0059$). **c, f** Isometric contractile properties of EDL muscles showing the peak tetanic force following 7 days (**c**) (150 Hz: Ctrl vs. Pred, $p = 0.3439$; Ctrl vs. RvD2, $p < 0.0001$; Pred vs. RvD2, $p < 0.0001$) or 21 days (**f**) of treatment (150 Hz: Ctrl vs. Pred, $p = 0.5265$; Ctrl vs. RvD2, $p = 0.0004$; Pred vs. RvD2, $p = 0.0027$) and **d, g** specific muscle force of EDL muscle of *mdx* mice treated for 7 days (**d**) (150 Hz: Ctrl vs. Pred, $p = 0.0634$; Ctrl vs. RvD2, $p < 0.0001$; Pred vs. RvD2, $p = 0.0027$), or 21 days (**g**) (150 Hz: Ctrl vs. Pred, $p = 0.8125$; Ctrl vs. RvD2, $p = 0.0003$; Pred vs. RvD2, $p = 0.0007$). **h** Timeline of weekly *ip* injection of either RvD2 (5 μg/kg), prednisone (2 mg/kg) or vehicle into *mdx* mice. **i, j** Isometric contractile properties of EDL muscles showing the peak tetanic force (**i**) (150 Hz: Ctrl vs. Pred, $p = 0.1908$; Ctrl vs. RvD2, $p = 0.0028$; Pred vs. RvD2, $p = 0.0697$), and specific force (**j**) (150 Hz: Ctrl vs. Pred, $p = 0.1537$; Ctrl vs. RvD2, $p = 0.0007$; Pred vs. RvD2, $p = 0.0318$) of *mdx* mice treated weekly for 21 days. **k** Timeline of weekly *ip* injection of O-1918 (Gpr18 antagonist; 2 mg/kg) alone or prior to the injection of RvD2 (5 μg/kg) or vehicle. **l, m** Isometric contractile properties of EDL muscles showing the peak tetanic force (**l**) (150 Hz: Ctrl vs. O-1918, $p = 0.2004$; Ctrl vs. RvD2 + O-1918, $p = 0.7759$; Ctrl vs. RvD2, $p < 0.0001$; RvD2 + O-1918 vs. RvD2, $p < 0.0001$), and specific force (**m**) (150 Hz: Ctrl vs. O-1918, $p = 0.7553$; Ctrl vs. RvD2 + O-1918, $p = 0.3862$; Ctrl vs. RvD2, $p < 0.0001$; RvD2 + O-1918 vs. RvD2, $p < 0.0001$) of *mdx* mice treated weekly for 21 days. **b–j** RvD2 = blue triangles, pred = red squares, and Ctrl = black circles. Data are presented as mean ± SEM, $n = 4$ (except $n = 3$ for **i, j, l,** and **m**) mice per group/time-point. Experiments were performed in technical duplicates using one-way (**b, e**) or two-way (**c, d, f, g, i, j, l,** and **m**) ANOVA uncorrected Fisher's LSD test. All data were analyzed with a 95% confidence interval. **$p < 0.01$; ****$p < 0.0001$ for **b** and **e**. *$p < 0.05$ compared with the vehicle, and †$p < 0.05$ compared with prednisone for **c, d, f, g, i,** and **j**. *$p < 0.05$ for RvD2 compared with either Ctrl or O-1918 or RvD2 + O-1918 (**l, m**).

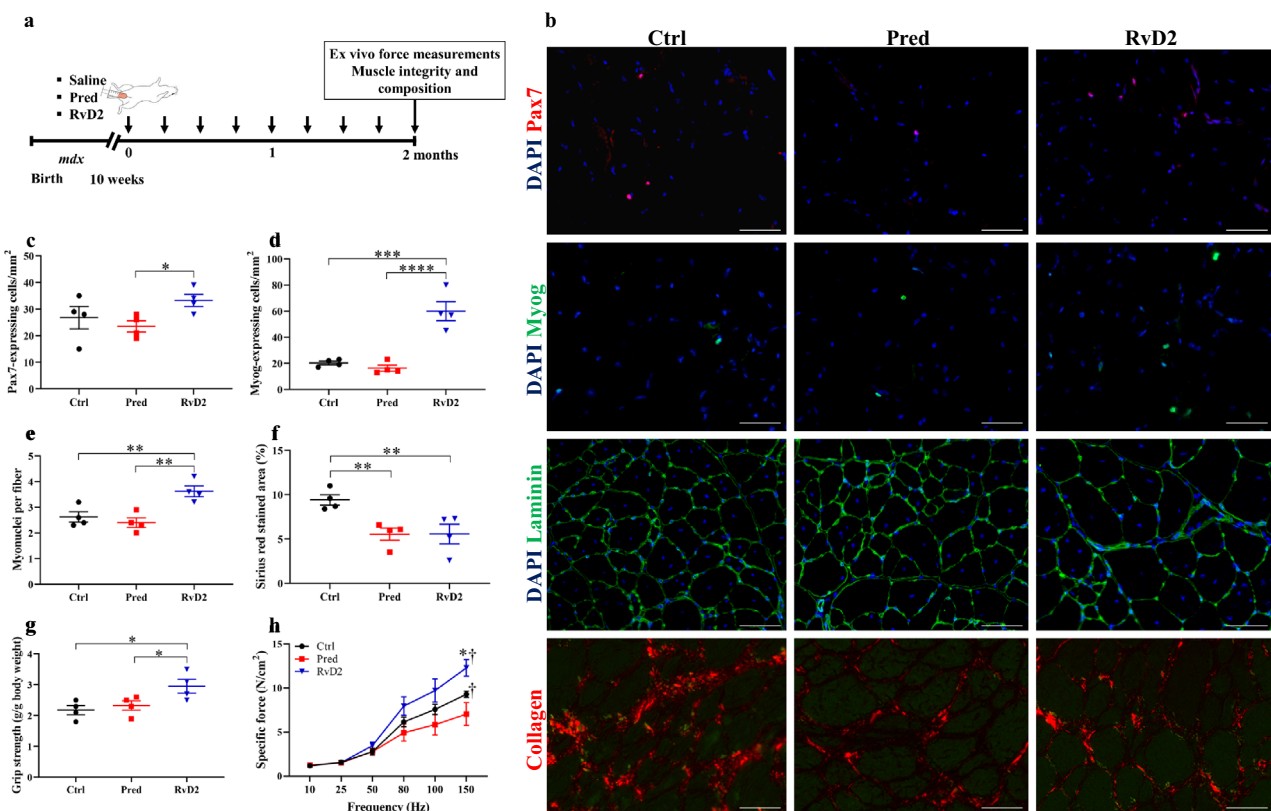

**Fig. 7 The superior therapeutic effect of RvD2 is maintained at long term in *mdx* mice. a** Timeline of weekly *ip* injection of either RvD2 (5 μg/kg), prednisone (pred, 2 mg/kg), or vehicle (ctrl) into *mdx* mice for 2 months. **b** Representative images of Pax7 (red), Myog (green), laminin (green), DAPI (blue), or collagen (red) immunohistological staining in muscle section of *tibialis anterior* (TA) muscle (for Pax7, Myog, laminin) or diaphragm muscle (for collagen staining) of treated *mdx* mice. Scale bars = 50 μm. **c, d** Quantification of total number of Pax7+ cells (**c**) (Ctrl vs. Pred, $p = 0.4663$; Ctrl vs. RvD2, $p = 0.1625$; Pred vs. RvD2, $p = 0.0484$) and Myog+ differentiated myoblasts (**d**) (Ctrl vs. Pred, $p = 0.5438$; Ctrl vs. RvD2, $p = 0.0001$; Pred vs. RvD2, $p < 0.0001$) in TA muscles of *mdx* mice treated for 2 months. **e** Number of myonuclei per fiber in TA muscles of *mdx* mice treated for 2 months (Ctrl vs. Pred, $p = 0.4460$; Ctrl vs. RvD2, $p = 0.0063$; Pred vs. RvD2, $p = 0.0019$). **f** Quantification of Sirius red positive area (collagen) in diaphragm muscles of *mdx* mice treated for 2 months (Ctrl vs. Pred, $p = 0.0089$; Ctrl vs. RvD2, $p = 0.0092$; Pred vs. RvD2, $p = 0.9834$). **g** Grip strength measured in treated *mdx* mice after 7 weeks of treatment (Ctrl vs. Pred, $p = 0.5667$; Ctrl vs. RvD2, $p = 0.0133$; Pred vs. RvD2, $p = 0.0351$). **h** Specific muscle force (N/cm$^2$) of EDL muscles of *mdx* mice treated for 2 months (150 Hz: Ctrl vs. Pred, $p = 0.0298$; Ctrl vs. RvD2, $p = 0.0048$; Pred vs. RvD2, $p < 0.0001$). **c–h** RvD2 = blue triangles, pred = red squares or Ctrl = black circles. Data are presented as mean ± SEM, $n = 4$ (except $n = 3$ for **h**) mice per group. Experiments were performed in technical duplicates using one-way (**c–g**) or two-way (**h**) ANOVA uncorrected Fisher's LSD test. All data were analyzed with a 95% confidence interval. *$p < 0.05$; **$p < 0.01$; ***$p < 0.001$; ****$p < 0.0001$ for **c–g**. *$p < 0.05$ compared with the vehicle; †$p < 0.05$ compared with prednisone for **h**.

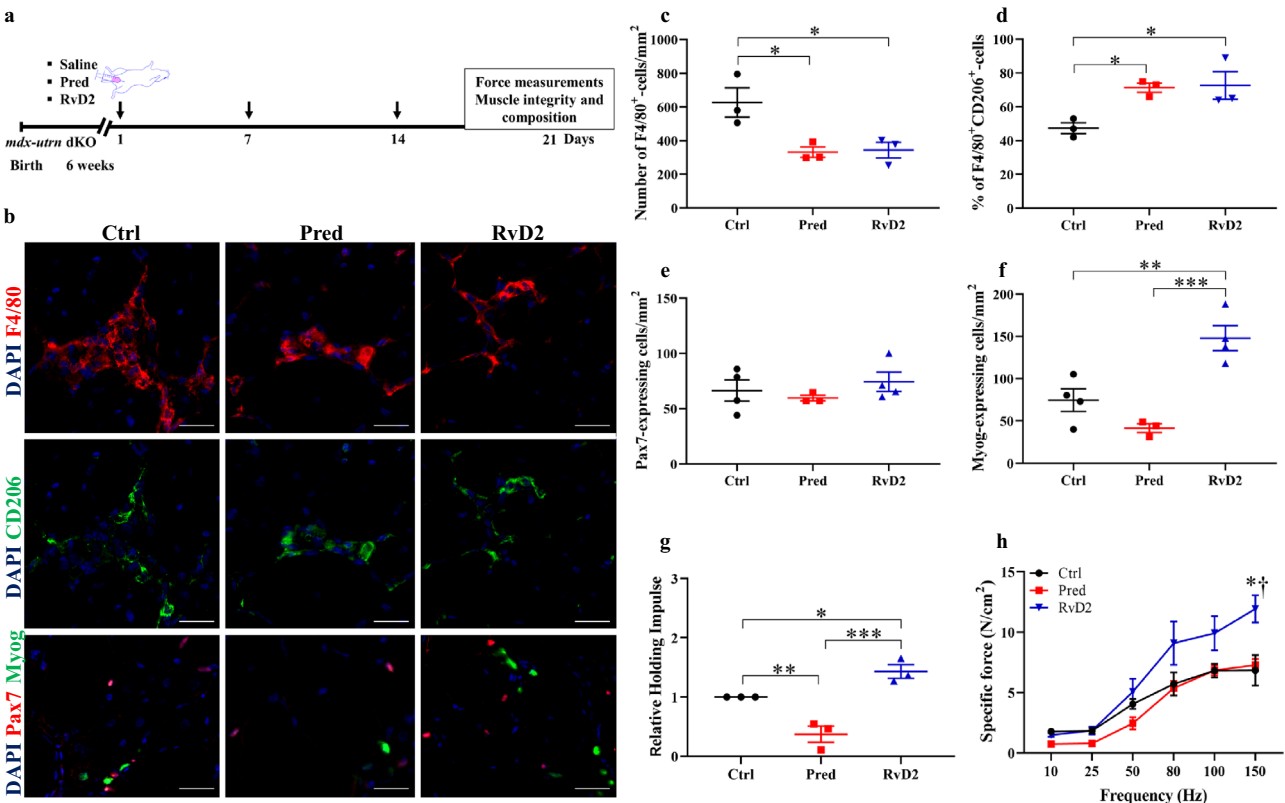

**Fig. 8 The superior therapeutic effect of RvD2 is maintained in the severely affected *mdx-utrn* dKO mouse model. a** Timeline of weekly *ip* injection of either RvD2 (5 µg/kg), prednisone (pred, 2 mg/kg) or vehicle (Ctrl) into *mdx-utrn* dKO mice for 3 weeks. **b** Representative images of F4/80 (red), and CD206 (green), or Pax7 (red) and Myog (green) immunostaining in muscle section of *tibialis anterior* (TA) muscle of treated *mdx-utrn* dKO mice. Scale bars = 50 µm. **c**, **d** Quantification of total number of F4/80$^+$ macrophages (**c**) (Ctrl vs. Pred, $p = 0.0126$; Ctrl vs. RvD2, $p = 0.0151$; Pred vs. RvD2, $p = 0.8851$), and proportion of anti-inflammatory macrophages (F4/80$^+$ CD206$^+$ / total F4/80$^+$) (**d**) (Ctrl vs. Pred, $p = 0.0186$; Ctrl vs. RvD2, $p = 0.0149$; Pred vs. RvD2, $p = 0.8647$) in TA muscles of *mdx-utrn* dKO mice. **e**, **f** Quantification of total number of Pax7$^+$ cells (**e**) and Myog$^+$ differentiated myoblasts (**f**) (Myog: Ctrl vs. Pred, $p = 0.1187$; Ctrl vs. RvD2, $p = 0.0031$; Pred vs. RvD2, $p = 0.0005$) in TA muscles of *mdx-utrn* dKO mice. **g** Hang test performance of *mdx-utrn* dKO mice treated for 3 weeks (Ctrl vs. Pred, $p = 0.0050$; Ctrl vs. RvD2, $p = 0.0248$; Pred vs. RvD2, $p = 0.0003$). **h** Specific muscle force (N/cm$^2$) of EDL muscles of *mdx-utrn* dKO mice treated for 3 weeks (150 Hz: Ctrl vs. Pred, $p = 0.7035$; Ctrl vs. RvD2, $p < 0.0001$; Pred vs. RvD2, $p = 0.0003$). **c–h** RvD2 = blue triangles, pred = red squares or Ctrl = black circles. Data are presented as mean ± SEM, $n = 3$ (except $n = 4$ for Ctrl and RvD2 in **e**, **f**) mice per group. Experiments were performed in technical duplicates using one-way (two-way for **h**) ANOVA uncorrected Fisher's LSD test. All data were analyzed with a 95% confidence interval. *$p < 0.05$; **$p < 0.01$; ***$p < 0.001$ for **c–g**. *$p < 0.05$ compared with vehicle, and †$p < 0.05$ compared with prednisone for **h**.

healthy regenerating muscles[59]. Even in dystrophic muscles, the presence of macrophages is essential to preserve the identity of MuSC and maintain their myogenesis potential[60]. Therefore, a timely and controlled balance in macrophage activity is required to optimize the function of MuSC in DMD. Our findings showed that RvD2 partially dampens inflammation but does not completely block leukocyte accumulation. Noteworthy, our findings showed that the capacity of RvD2 to resolve inflammation in DMD is similar after systemic (i.p.) or local delivery (i.m.), suggesting that its action is independent of the administration route and acts locally in skeletal muscles. This capacity of Resolvins to promote the active resolution of inflammation has been observed in other pathological conditions such as asthma, arthritis, sepsis, among others[23–27,61,62]. Similar to what was observed in acute muscle injury (cardiotoxin)[29], our results indicated that RvD2 promotes the switch of macrophages toward their anti-inflammatory phenotype, which is known to contribute to myoblast differentiation/fusion and improve muscle regeneration[59,60,63]. Our in vitro data showed that RvD2 reduced the expression of *Tnfa*, *Cxcl1*, and *Tgfb* by non-polarized macrophages, which are known to inhibit myoblast differentiation and fusion[59,64]; and tended to increase the expression of *Il-4*, an anti-

inflammatory cytokine that promotes myoblast fusion[65]. Moreover, RvD2 decreased the expression of genes involved in the production of prostaglandins (*Ptgs2*, *Ptges*) and increased the expression of Annexin-A1 (*Anxa1*), a potent indirect inhibitor of eicosanoids production, in M1-polarized macrophages[66]. Considering our observation that the prostaglandins/resolvins ratio is increased in the muscles of *mdx* mice, the administration of RvD2 could restore the balance in pro-inflammatory/pro-resolving signals, not only by providing an external source of Resolvins but also by intrinsically reducing the endogenous production of prostaglandins by inflammatory cells. Overall, RvD2 acts through macrophages and promotes their anti-inflammatory/pro-regenerative phenotype to indirectly restore the myogenesis capacity of MuSC, in addition to their direct effects on myoblasts.

The lack of dystrophin in DMD also intrinsically affects the myogenesis capacity of MuSC. Different studies in mice, zebrafish, and humans, have shown that the dystrophin protein is transiently expressed in MuSC[14,67–69]. In dystrophin-deficient MuSC, the lack of dystrophin impairs cell polarity establishment and asymmetric cell division, leading to a reduction in the production of differentiated myoblasts and impaired muscle regeneration[14]. Dystrophin-deficient myoblasts cultured in vitro

also display a reduced proliferation, differentiation, and fusion capacity[12,13,70]. These defects are rescued when dystrophin expression is restored by CRISPR-Cas9, indicating that dystrophin-deficient MuSC is intrinsically compromised in their ability to generate committed myoblasts[11]. Notably, treatment with small molecules, such as Epidermal Growth Factor (EGF), was shown to rescue the defects in dystrophin-deficient MuSC and to promote the generation of differentiated progenitors leading to enhanced muscle function[15]. Thus, there is a high therapeutic potential for molecules aiming to restore the differentiation potential of MuSC and enhance their myogenesis capacity[41]. Our findings showed that RvD2-driven muscle regeneration is not only attributable to improved resolution of inflammation but mostly relies on its capacity to rescue myogenic cell function and myogenesis. Our data using *mdx* mice indicates that RvD2 increases the total number of myogenic cells, the number of myonuclei/fiber, and the proportion of MyHC-emb+ regenerating myofibers, but does not significantly reduce the number of IgG-positive necrotic myofibers, suggesting that the therapeutic effect of RvD2 is mediated through improved myogenesis rather than reduced degeneration. Particularly, we showed that, in addition to stimulating the release of myogenic factors by macrophages, RvD2 is also able to directly target myogenic cells to promote their differentiation. This effect is mediated by the receptor Gpr18 expressed by myogenic cells. RvD2 treatment activates the Akt pathway in myogenic cells, which is known to promote myoblast differentiation and fusion[71,72]. The activation of the RvD2-Gpr18-Akt-mTOR pathway has also been observed in other cell types such as endothelial cells and cardiomyocytes[73,74]. Similar to RvD2, different specialized pro-resolving mediators were shown to directly affect the function of other stem cell types[38]. For instance, the substrate of RvD2, docosahexaenoic acid, promotes the differentiation of neural stem cells by regulating basic Helix-loop-Helix transcription factors and cell cycle exit[75]. Similarly, neuroprotectin-D1 promotes neuronal and cardiac differentiation of embryonic stem cells[76]. Lipoxin-A4 derived from the metabolism of arachidonic acid through lipoxygenases can directly target periodontal stem cells to promote their wound healing capacity[77]. Our findings show that RvD2 can directly target myogenic cells, which adds to the growing evidence that specialized pro-resolving mediators regulate stem cell biology.

We also observed that FAPs (muscle resident mesenchymal stem cells) express Gpr18, albeit at much lower than proliferating myoblasts (which already express a lower level of Gpr18 compared to differentiated myoblasts). Although we did not observe a direct impact on FAPs proliferation in vitro, further study will be needed to clearly determine the effect of RvD2 on this stem cell population. Importantly, FAPs are the main cellular source of fibrosis in dystrophic skeletal muscle[78], and our in vivo results show that RvD2 decreases fibrosis to the same level as prednisone. Therefore, considering that the anti-fibrosis and anti-inflammatory capacities of RvD2 and prednisone are similar, the only difference that explains the superior therapeutic impact of RvD2 on muscle function is its impact on myogenic cells.

The balance between self-renewal and differentiation of MuSC is critical to ensure the long-term regenerative potential of skeletal muscles. Noteworthy, both our ex vivo (single myofiber culture) and in vivo experiments indicated that RvD2 promotes the generation of differentiated myoblasts without reducing the absolute number of Pax7+ MuSC. The effect of RvD2 on myoblast differentiation is less pronounced in ex vivo single muscle fibers than in primary myoblasts in vitro, which suggests that RvD2 preferentially targets myogenic cells at the progenitor stage rather than quiescent/activating MuSC. The superior impact of RvD2 on myogenic progenitors is consistent with the increased expression

of Gpr18 during myoblast differentiation. These findings explain that RvD2 does not promote the exhaustion of the MuSC pool in vivo, even at a longer term (2 months of treatment). These data suggest that RvD2 does not overdrive MuSC toward differentiation but is rather restoring the cell differentiation potential and generation of committed progenitor cells, which is defective in dystrophin-deficient myogenic cells[11,14,15,70].

In summary, the pathogenesis of DMD is a complex process, which is caused by a variety of factors such as myofiber necrosis, excessive and disorganized inflammation, fibrofatty tissue replacement, and impaired regenerative capacity of MuSC. Glucocorticoids are able to reduce chronic inflammation, but they directly stimulate muscle fiber atrophy and impair MuSC functioning[19–21]. On the other hand, RvD2 represents a versatile therapeutic option that targets many of the harmful effects caused by the lack of dystrophin. The findings of this preclinical study suggest that RvD2 has a higher therapeutic potential to mitigate DMD compared to the current gold-standard treatment. Despite hundreds of clinical trials on DMD, glucocorticoids remain the only drug that consistently showed an improvement in the preservation of muscle force and function in patients[16], suggesting that targeting inflammation remains a viable therapeutic avenue that can lead to clinical benefits. Overall, this preclinical study used standardized operating procedures, meaningful outcomes, different mouse models of DMD, and comparison to a gold standard to demonstrate the potent therapeutic potential of RvD2. These results are particularly important considering that Resolvins and specialized pro-resolving mediators are currently being tested in clinical trials and have shown their safety and their promising therapeutic potential for acute and chronic pathologies[79–81].

## Methods

**Animals and experimental design**. Ten-week-old male *mdx* mice (Jackson Laboratory; Bar Harbor, ME) were housed on a 12:12 h light:dark cycle at 21 °C and 40% humidity in pathogen-free cages within the Animal Holding Facility at the CHU Sainte-Justine (Montreal, CA). Mice had free access to food and tap water and were randomly assigned to three experimental groups that received either daily or weekly intraperitoneal (i.p) injection of saline (control), RvD2 (5 µg/kg/day[24,29,82,83]), or prednisone (2 mg/kg/day[45]). In another set of experiments in which the molecular mechanism underlying the beneficial effect of RvD2 on dystrophic muscles was investigated, the mice were also pretreated with O-1819, a Gpr18 antagonist that has been previously used in vivo to block RvD2-induced Gpr18 signaling[47,50,51]. Ten-week *mdx* mice received a weekly i.p injection of ether saline, O-1918 (2 mg/kg), RvD2 (5 µg/kg), or O-1918 (30 min prior RvD2 administration) + RvD2. Global physical function was recorded using validated in situ procedures. After 7 or 21 days of treatment, the *tibialis anterior* (TA) muscle, the *extensor digitorum longus* (EDL), and the diaphragm were collected, weighed, and processed for further analyses. The EDL was used to assess ex vivo muscle function. The TA and diaphragm muscles were embedded in freezing medium (OCT) and rapidly frozen in liquid nitrogen-cooled isopentane and stored at −80 °C prior to sectioning and staining. For experiments carried out on utrophin-dystrophin double knockout mice (*mdx-utrn*-dKO), the mice were used at 6 week-of-age and received weekly i.p injection of either saline, prednisone, or RvD2 for 3 weeks. Mice were euthanized by $CO_2$ inhalation (under anesthesia) followed by cervical dislocation. All animal experiments were approved by the CHU Sainte-Justine Research Ethics Committee and performed in compliance with the Comité Institutionnel des Bonnes Pratiques Animales en Recherche (CIBPAR; approval number 2020-2668) in accordance with the Canadian Council on Animal Care guidelines.

**Four limb hanging test**. This non-invasive method is used to evaluate global physical function in mouse models of neuromuscular diseases[84]. We performed the measurements according to the standard operating procedures established by TREAT-NMD (protocol DMD_M.2.1.005)[84]. Briefly, mice were placed individually inside a 35 cm wire grid cylinder set on the top of a mouse cage containing 6 cm of soft bedding. After five seconds of accommodation, the cylinder was inverted and the hang time (defined as the amount of time it takes the mouse to fall) was recorded. The capacity of the mouse to oppose the gravitational force, referred to as the holding impulse, was calculated by multiplying the body weight by the hang time. Each mouse was tested 3 times (2 min rest between trials), and the best trial was taken.

**Grip strength**. The forelimb grip strength was measured on *mdx* mice according to the DMD_M.2.2.001 protocol described by TREAT-NMD[85]. Briefly, mice were suspended by their tail and allowed to grasp with their forelimb a horizontal metal bar attached to an automated grip strength meter (Bioseb, FL, USA). Once both paws were symmetrically aligned on the bar, the mouse body was horizontally maintained, and gently pulled away by the tail until breaking the grasp. The test was repeated 3 times at 1 min interval, and the best trial was used for analysis. The grip strength meter was reset to zero before each measurement. Results were normalized to body weight.

**Ex vivo muscle contractility**. Isometric contractile properties were assessed as previously described[86,87], according to the standard operating procedures described by TREAT-NMD (protocol DMD_M.1.2.002)[88]. Prior the measurements of muscle contractility, mice received a dose of buprenorphine (0.1 mg/kg, i.p.) and 50 mg/kg of pentobarbital sodium. Then, the left EDL was carefully isolated and placed in a buffered physiological solution (Krebs-Ringer supplemented with glucose and bubbling carbogen gas) maintained at 25 °C. The muscles were attached by the tendons to an electrode at one end and to a lever arm at the other end (300C-LR dual-mode lever; Aurora Scientific, Canada). Optimal muscle length ($L_0$) was determined and gradually adjusted until the maximum isometric twitch tension was achieved. At which point the muscles were allowed a 10 min equilibration in the bath prior to the measurement of contractile properties. A force-frequency curve was established by stimulating the muscles at different frequencies, ranging from 10–150 Hz, with a 2 min rest between each stimulation. Thereafter, muscle length and weight were measured to assess specific muscle force (N/cm$^2$). To do so, a mathematical approximation of the cross-sectional area was calculated based on the formula: Specific force (N/cm$^2$) = (absolute force (N) × fiber length (0.44 × $L_0$ for the EDL muscle) × muscle density (1.06 g/cm$^3$))/muscle mass (g).

**Single myofiber isolation and culture**. Single myofibers were isolated from the EDL of euthanized *mdx* mice[14,89]. Both EDL were carefully dissected and digested at 37 °C in DMEM (Gibco) with 0.2% collagenase type I (Worthington Biochemical Corporation) solution for 45 min. Single myofibers were isolated with a glass pipette by gently triturating the digested muscles. Myofibers were cultured in DMEM supplemented with 20% FBS (Wisent), 1% penicillin–streptomycin, 1% chicken embryo extract, with or without prednisone (10 μM) or RvD2 (200 nM). Fibers were collected after 72 h of culture, and immunofluorescence was performed to determine the number of Pax7 and myogenin-expressing cells.

**Monocytes/macrophages isolation and culture**. The femur and tibia were dissected intact, cleaned, and the bone marrow was flushed out with medium using a 25 G needle. Monocytes were purified by magnetic-activated cell sorting using the Monocyte Isolation Kit (Miltenyi Biotec, Germany) and differentiated into macrophages[87]. Briefly, monocytes were cultured for 4 days in a macrophage medium containing DMEM GlutaMax high glucose, 1X pyruvate, 10% FBS, 1% penicillin–streptomycin, and 10 ng/ml macrophage colony-stimulating factor. Macrophages were then stimulated for 24 h with LPS (100 ng/ml) and IFN-γ (20 ng/ml) to induce M1-polarization[35,90]. Thereafter, macrophages were incubated for 48 h in a serum-free macrophage medium supplemented with or without RvD2 (200 nM; Cayman Chemical, USA). The macrophage-conditioned medium was collected, centrifuged, and added to the primary myoblast culture for 3 days to assess the impact of macrophage-secreted paracrine factors on myoblast differentiation/fusion. To assess the impact on myoblast proliferation, the macrophage-conditioned medium (2:1 in proliferating medium) was incubated with myoblasts overnight (16 h), and Ki67 immunostaining was performed to assess the proportion of proliferative cells.

**Flow cytometry**. The phenotype of M1-polarized macrophages treated with RvD2 or vehicle was further analyzed by flow cytometry, using a sequential of extracellular and intracellular staining. Single-cell suspensions were washed and incubated for 30 min at 4 °C with the eBioscience Fixable Live–Dead Viability Dye Fluor 506 (Ottawa, ON, Canada). Cell suspensions were washed, incubated with 10% FcR block for 15 min, and labeled for 30 min at 4 °C with F4/80-FITC conjugated diluted at 1:50 (clone REA126, Miltenyi Biotec, California, USA). Samples were washed, fixed with 2% PFA (in PBS 1X) for 5 min, and permeabilized with 0.2% Triton X-100 for 10 min. Thereafter, samples were incubated for 30 min at 4 °C with the intracellular iNOS-PE-conjugated anti-mouse antibody (clone REA982, 1:50 in permeabilization buffer, Miltenyi Biotec, California, USA). After washing, cell profile was determined using an LSR Fortessa flow cytometer (5 L B/R/V/UV, Ottawa, Ontario, Canada). Data were analyzed using FlowJo v10.6.2 software (FlowJo, LLC).

**ELISA**. The level of RvD2 was quantified in the macrophage medium using the Cayman's Resolvin D2 ELISA kit (Product number: 501120) according to the manufacturer's protocol. This kit is a competition-based assay between free RvD2 and RvD2-linked to acetylcholinesterase (RvD2 tracer) for a limited number of specific rabbit antisera binding sites. The level of RvD2 in the sample is inversely correlated to the amount of RvD2 tracer bound to the rabbit antiserums. Samples were developed with Ellman's reagent and detected at 420 nm (CLARIOstar

Mandel, BMG Labtech, Ontario, Canada). Data were expressed as a percentage of the initial amount of RvD2 (200 nM) added to the medium.

**MuSC isolation and primary myoblast culture**. Muscles from both hindlimbs of *mdx* mice were collected and dissociated in collagenase/dispase (Sigma) using the gentle MACS dissociator (Miltenyi Biotech), filtered through a 30 μm cell strainer, and stained with antibodies for 30 min on ice in the dark[14]. MuSC were isolated by FACS using a gating strategy based on forward scatter and side scatter profiles, cell viability (7-AAD; 1:40; Biolegend), negative selection with FITC-conjugated antibodies for anti-Sca-1 (Clone D7; 1:30; Miltenyi Biotech), anti-CD45 (clone 30F11; 1:30; Miltenyi Biotech), anti-CD31 (Clone 390; 1:30; Miltenyi Biotech), anti-CD11b (M1/70.15.11.5; 1:30; Miltenyi Biotech), and positive selection for APC-conjugated anti-Itgb1 (clone HMβ1-1; 1:15; Miltenyi Biotech), and PE-conjugated anti-Itga7 (clone 3C12, 1:100; Miltenyi Biotech) (Supplementary Fig. 12a)[14]. Isolated cells were cultured in collagen-coated Petri dishes with Ham's F10 media (GIBCO) containing 20% FBS (Wisent), 1% penicillin–streptomycin (GIBCO), and 2.5 ng/mL bFGF (Wisent). Cells were treated with or without prednisone (10 μM) or RvD2 (200 nM) and incubated in the IncuCyte (Essen Bioscience), an automated microscope located in the cell culture incubator that automatically takes pictures and performs unbiased analysis of cell numbers. Myoblasts were incubated in a low-serum differentiation medium (50% Ham's F10, 50% DMEM low glucose, 1% penicillin–streptomycin, 5% horse serum) supplemented or not with prednisone (10 μM) or RvD2 (200 nM) for 16 h (overnight) to assess myoblast differentiation or 4 days for myoblast fusion and myotube formation. Immunofluorescence was performed at the end of the experimental procedures. Fusion index was determined as the proportion of total nuclei located in multinucleated myotubes.

**FAPs isolation and culture**. Muscle tissues were dissociated as described above, and FAPs were sorted using a gating strategy based on forward scatter and side scatter profiles, cell viability (7-AAD; 1:40; Biolegend), negative selection with FITC-conjugated antibodies for anti-CD45 (clone 30F11; 1:30; Miltenyi Biotech), anti-CD31 (Clone 390; 1:30; Miltenyi Biotech), anti-CD11b (M1/70.15.11.5; 1:30; Miltenyi Biotech), PE-conjugated anti-Itga7 (clone 3C12; 1:100; Miltenyi Biotech) and positive selection for APC-Vio 770 anti-CD140a (Clone REA637; 1:15; Miltenyi Biotech), and BV421 anti-Sca-1 (Clone D7; 1:100; Biolegend)(supplementary. Fig. 12b). Isolated cells were cultured with DMEM (GIBCO) containing 10% FBS (Wisent), 1% penicillin–streptomycin (GIBCO), and 2.5 ng/mL bFGF (Wisent).

**siRNA transfection**. Transient knockdown of Gpr18 was performed using a stealth siRNA duplex construct (MSS201297, Thermo Fisher) on primary *mdx* myoblasts. Stealth siRNA duplexes at a concentration of 48 nM were transiently transfected into cells using Lipofectamine RNAiMAX (Thermo Fisher), according to the manufacturer's specifications. Cell media was changed to differentiation media 24 h after siRNA transfection, and cells were differentiated with or without RvD2 for 4 days.

**Histology**. Hematoxylin and Eosin (H&E) staining was performed for the histological characterization of dystrophic muscles according to standard operating procedures described by TREAT-NMD (MDC1A_M.1.2.004)[91]. Briefly, 10 μm thick sections were cut from the proximal and distal half of the TA muscle, mounted on Superfrost Plus slides (Thermo Fisher Scientific, Canada), and fixed with paraformaldehyde (2% PFA). Sections were washed and incubated with Mayer's hematoxylin solution for 5 min, following a 10 min wash in a glass chamber under warm running tap water. Slides were incubated for 10 min with acetic acid 0.5% Eosin solution (Merck 1.09844.1000), washed three times with bidistilled water, dehydrated by consecutive dipping in 70%, 90%, and 100% ethanol and xylene. Coverslips were mounted onto slides using the Eukitt quick-hardening mounting medium (Sigma-Aldrich, Oakville, Ontario, Canada). The entire sections were captured at 200x magnification using the digital microscope slide scanner (Zeiss AxioScan.Z1). Images were then analyzed blindly using the ImageJ Analysis Software Program (version 1.53, National Institutes of Health, Maryland, USA) to measure the minimal fiber diameter. The fibers of the entire muscle section were measured.

To assess fibrosis, muscle sections from *mdx* diaphragm were stained using the Picrosirius red kit (Polysciences, Inc, Pennsylvania, USA). Briefly, sections were fixed with 4% PFA for 5 min, rinsed in distilled water, and incubated with the Picrosirius red solution for 90 min at room temperature. Sections were rinsed with hydrochloride acid (1 N) and distilled water before a full step of dehydration as described above. Sections were mounted and visualized on a Leica microscope (Leica Microsystems, Canada) under polarized light. The proportion of area stained in red (collagen) relative to the entire muscle section was quantified with the ImageJ software[92].

**Immunofluorescence**. Immunodetection was performed on cultured cells or on skeletal muscle sections[93]. Briefly, samples were fixed with 2% PFA for 5 min, quenched with 0.1% glycine solution for 5 min, and permeabilized with 0.2% Triton X-100 for 10 min. Samples were blocked with mouse on mouse blocking reagent (Vector labs), 5% goat serum, and 2% BSA in PBS for 90 min at room temperature and incubated overnight at 4 °C with the following primary antibodies:

Mouse anti-Pax7 (clone PAX7, 1:20; Developmental Studies Hybridoma Bank (DSHB), created by the NICHD of the NIH and maintained at The University of Iowa), rabbit anti-myogenin (clone EPR4789; 1:500; Abcam), mouse anti-Myh3 (clone F1.652; 0.3 µg/ml; DSHB), rabbit anti-Laminin (cat# ab11575; 1:1,000; Abcam), rat anti-F4/80 monoclonal antibody (clone A3-1; 1:1,000; Bio-Rad), rabbit CD206 monoclonal antibody (cat# ab64693; 1:2,000; Abcam), rat anti-Ly-6G monoclonal antibody (clone 1A8; 1:250; Invitrogen), rabbit CD3 monoclonal antibody (clone SP7; 1:1,000; Abcam), mouse anti-myosin heavy chain (clone MF20; 1:20; DSHB), rat anti-mouse CD25-PE conjugated (Clone PC61; 1:40; BD Bioscience), F4/80-FITC conjugated (clone REA126; 1:50; Miltenyi Biotec), rat anti-mouse CD163-PE conjugated (clone TNKUPJ; 1:50; Thermo Fisher Scientific), rat anti-mouse arginase-APC conjugated (clone A1exF5; 1:50; Thermo Fisher Scientific), rat Ki67 monoclonal antibody (clone SolA15, 5 µg/ml, Thermo Fisher Scientific), or rabbit polyclonal anti-Gpr18 (1:1,000; Sigma). Antibodies were validated using positive and negative controls. Samples were washed with PBS and incubated with appropriate cross-adsorbed secondary antibodies (Invitrogen, Thermo Fisher), consisting of either goat anti-rabbit IgG H + L (Alexa Fluor 488, 1:1,000), goat anti-mouse IgG1 (Alexa Fluor 546, 1:1,000), goat anti-rat IgG H + L (Alexa Fluor 594, 1:1,000), or goat anti-mouse IgG H + L (Alexa Fluor 488, 1:1,000) for 1 h at room temperature. Samples were washed with PBS and counterstained with DAPI. Sections were then mounted with PermaFluor (Thermo Fisher Scientific). Images were viewed and captured on the entire sections using an epifluorescence Leica DM5000 B (Leica Microsystems, Canada) or EVOS M5000 (Thermo Fisher Scientific).

**Western blot**. Cells were washed with sterile PBS and lysed with ice-cold RIPA buffer containing 1% of protease inhibitors and centrifuged at 10,000 g for 10 min. The supernatant was retained, aliquoted, and the protein content was quantified using the BCA Assay Kit (Thermo scientific, Mississauga, Ontario, Canada). A volume corresponding to 40 µg of protein was diluted with a sample buffer (125 mM Tris buffer (pH6.8), 4% SDS, 20% glycerol, 0.05% bromophenol blue, and 200 mM dithiothreitol), heated at 100 °C for 5 min and electroseparated on 8 or 10% sodium dodecyl sulfate-polyacrylamide gel. Proteins were transferred to polyvinylidene difluoride membranes, which were blocked with 5% non-fat milk or 5% BSA for 90 min at room temperature, and immunoblotted overnight at 4 °C with either mouse anti-myosin heavy chain (clone MF20; 1:20 in 5% non-fat milk; DSHB), rabbit anti-Gpr18 (cat# SAB4501253; 1:1,000 in 5% non-fat milk; Sigma), rabbit anti-(pan)-Akt (clone C67E7; 1:1,000 in 5% BSA; Cell Signaling Technology), rabbit anti-phospho-Akt (Ser$^{473}$) (cat# 9271; 1:1,000 in 5% BSA; Cell Signaling Technology), rabbit anti-β-actin (cat# 4967; 1:1,000 in 5% non-fat milk; Cell Signaling Technology), or rabbit anti-GAPDH (cat# 2118; 1:1,000 in 5% non-fat milk; Cell Signaling Technology) as primary antibodies. After washing, membranes were incubated with goat anti-rabbit (H + L) or goat anti-mouse (H + L) HRP-conjugated secondary antibodies (1:3,000; Abcam) for 1 h at room temperature. Bands were revealed with ECL-plus Western blotting reagent (PerkinElmer Life and Analytical Sciences, USA), visualized with the GeneSys image software (Syngene), and quantified using ImageJ (National Institutes of Health, Maryland, USA). Densities were normalized to the loading control. Uncropped and unprocessed scans are shown in Supplementary Fig. 13.

**qPCR**. Macrophage total RNA was extracted with Qiazol reagent according to the manufacturer's specifications. Total RNA was quantified with a Thermo Scientific™ NanoDrop™ 8000 Spectrophotometer. The reverse transcription was performed on 1.5 µg of total RNA using the 5X All-In-One RT MasterMix (Abcam, G486) to obtain cDNA. qPCR was performed with a set of primers designed on Primer-BLAST (NCBI) and validated for their specificity, efficiency, and annealing temperature[94]. Gene amplification was performed with the BrightGreen 2X qPCR (Abcam, Mastermix-s-xl) on a Roche LightCycler® 480 Instrument II. Data were analyzed with the LightCycler® 480 and were normalized relative to *GAPDH* expression. The primers used are shown in Supplemental Table 1.

**Mass spectrometry (HPLC-MS/MS)**. Resolvin (Rv) D1, RvD2, RvD3, 17(R, S)-RvD4, RvD5, RvE1, Prostaglandin (PG) E₂, PGF$_{2α}$, Leukotriene (LT)-B₄, and thromboxane (Tx)-B₂, were purchased from Cayman Chemical (Ann Arbor, MI, USA). Butylated hydroxytoluene (BHT) was acquired from Sigma-Aldrich (Oakville, ON, Canada). Sodium acetate trihydrate (ACS grade) was obtained from Laboratoire Mat (Québec, QC, Canada). N-hexane 95% (OPTIMA), Acetonitrile (HPLC grade), Methanol (HPLC grade), and indomethacin were bought from Fisher Scientific (Ottawa, ON, Canada). Potassium hydroxide, ammonium hydroxide, and acetic acid were purchased from VWR International (Ville Mont-Royal, QC, Canada) and ethanol 99% from Commercial Alcohols (Toronto, ON, Canada).

Gastrocnemius muscles (120 mg) in 1 mL of methanol containing 0.04% BHT and 2 µM indomethacin was homogenized with a bead-beating homogenizer system (FastPrep-24 from MP Biomedicals). Briefly, 3 cycles of 30 s at 10.0 m/sec were performed in 2 mL tube containing ceramics beads of 2.8 mm (Omni International, Kennesaw, GA, USA). Tubes were cooled down in a benchtop cooler between each cycle. Homogenates corresponding to ~ 50 mg of tissues were mixed with 2.5 mL of acetate buffer pH 3 and 10 µL of deuterated oxylipin internal standards (25 ng/mL in ethanol). Samples were centrifuged for 5 min at 4,000 × g

before solid-phase extractions (SPE). SPE cartridges (Oasis MAX 60 mg/3cc, Waters Corporation Mississauga, Ontario, Canada) were conditioned with methanol and Milli-Q water. Washing steps were made by successive additions of sodium acetate buffer (50 mM, pH 3), N-hexane, ammonium hydroxide (2.5 mM), and lastly with acetonitrile–methanol mixture (8:2 v/v). Oxylipins were then eluted with a mixture of 79.2% of acetonitrile, 19.8% methanol, and 1% acetic acid and dried under a stream of nitrogen. Samples were reconstituted with 60 µL of 30% acetonitrile-0.01% acetic acid and filtered with Nanoseps centrifugal devices from Pall Corporation (0.2 µm wwPTFE; Pall Biotech—Canada, Mississauga, Ontario, Canada) prior to HPLC injection. The same procedure was used to build a spiked calibration curve.

Oxylipin quantification was performed as described previously[95]. Acquisition was done through Analyst 1.7 (AB Sciex, Framingham MA, USA) and the mass spectrometer was operated in multiple-reaction monitoring mode according to transitions shown in Supplemental Table 2. The quantification was achieved by using SCIEX OS 1.7MQ software (AB Sciex).

**Statistics and reproducibility**. Sample size determination was based on the expected effect size and variability that was previously observed by the investigators for similar readouts using *mdx* mice[14,15]. No data were excluded. All experiments were repeated independently at least twice in the laboratory with similar results. For data collection and analysis, the experimenter was blinded to the identity of the sample. Data were analyzed with the MIXED procedure of the Statistical Analysis System (SAS Institute, version 9.2, Cary, North Carolina, USA). Normality was verified for all data according to the Shapiro–Wilk test. Treatment effects were determined with two-tailed Student's *t*-tests, One-way or Two-way analysis of variance (ANOVA) uncorrected Fisher's Least Significant Difference (LSD) test. Results are reported as mean ± standard error of the mean (SEM).

**Reporting summary**. Further information on research design is available in the Nature Research Reporting Summary linked to this article.

## Data availability
The raw data generated for all figures (Figs. 1–8 and Supplementary Fig. 1–11) of this study are provided in a Supplementary Excel file. Source data are provided with this paper.

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

## Acknowledgements

Samples from Gpr18-knockout mice were kindly provided by Dr. Jason Cyster (University of California San Francisco). We thank M. Kevin Amélété, Ms. Marie-Michèle Forget, Ms. Léa Dorion, and Ms. Catherine Aaron for their technical assistance. J.D. was supported by fellowships from the Sainte-Justine Foundation, the Association Française contre les Myopathies (AFM-Telethon), and the Muscular Dystrophy Association. P.F. was supported by awards from the Sainte-Justine Foundation, CERMO-FC, and FRQS doctoral awards (Fonds de recherche du Québec – Santé). T.M. was supported by the Sainte-Justine Foundation and FRQS doctoral awards. T.C.C. was supported by a fellowship from the Myotonic Dystrophy Foundation. Z.O. was supported by an award from the Burroughs Wellcome Fund and MITACS fellowship. N.A.D. was supported by the FRQS Junior-1 award. This study was supported by grants from the Canadian Institutes of Health Research, Natural Sciences and Engineering Research Council of Canada, Canada Foundation for Innovation, Stem Cell Network, the CHU Sainte-Justine Foundation, the Grand Défi Pierre Lavoie Foundation, and the Quebec Cell, Tissue and Gene Therapy Network –ThéCell (a thematic network supported by the FRQS).

## Author contributions

N.A.D. conceived and managed the project. J.D., P.F., T.M., Z.O., T.C.C., K.G., O.P., J.F.B and N.A.D. designed, performed experiments, and analyzed the data. J.D. and N.A.D. interpreted the data and wrote the manuscript. All authors carefully reviewed and provided critical insights into the manuscript.

## Competing interests

The authors declare no competing interests.
