## [Peer Review File · Nature Communications]

Reviewers' Comments:

Reviewer #1:

Remarks to the Author:

The present work by Dort et al investigates the role of Resolvin-D2 (RvD2) in Duchenne Muscular Dystrophy (DMD). RvD2 has been previously shown to have anti-inflammatory function and to be transiently upregulated during acute skeletal muscle injury, and the current manuscript explores its role in DMD and compare RvD2 treatment with standard of care glucocorticoids in DMD mouse models. The authors show that macrophages isolated from mdx mice and treated with RvD2 transition to an anti-inflammatory phenotype, expressing CD206 and Il-4 and downregulating inflammatory signals such as Tnf, Tgfbeta and Cxcl1. When mdx primary myoblasts were exposed to conditioned media from mdx macrophages treated with RvD2, they exhibited increased fusogenic capacity compared to control untreated macrophages. The authors further show that RvD2 has a direct effect on muscle stem cells, as when added to the media of freshly isolated muscle stem cells or isolated myofibers, they observe an increase in differentiation. They evaluated the expression of Grp18, a cell surface receptor previously shown to mediate the effects of RvD2 in inflammatory cells. The authors show that this receptor is expressed in mdx myoblasts and that its downregulation prevents the effect of RvD2 on myoblast differentiation and fusion. Upon treatment with RvD2, the work provides evidence that muscle cells upregulates the Akt pathway. The authors further evaluated the effects of RvD2 treatment in mdx mice and compared it to prednisolone treatment. RvD2 had comparable effects on inhibiting inflammation in dystrophic muscles. Interestingly, RvD2 treatment, in contrast to prednisone, had a significant effect in increasing muscle stem cell differentiation in vivo, as shown by an increase in the number of myogenin+ cells, in regenerating myofibers and myofiber cross-sectional area. Finally, the authors show that RvD2 treatment improves muscle function in two different DMD models. The development of novel therapeutic approaches for DMD is a major clinical need, thus the findings reported are highly relevant. The work identified RvD2 as a mediator that has the dual effect of inhibiting inflammation and at the same time promoting myogenic differentiation of muscle stem cells, resulting in improved tissue form and function. The findings are timely and novel, the experiments are rigorously performed, the manuscript is well written and the data clearly presented. However, additional experiments are required to fully support the authors' interpretation.

Main points:

1- Conclusive evidence of binding of RvD2 to Grp18 in muscle cells: While the authors show that acute downregulation or genetic deletion of Grp18 prevents the effect of RvD2 on mdx myoblasts on differentiation/fusion and Akt activation, a conclusive demonstration that it is mediated by direct binding in muscle cells is not provided. Inclusion of mass-spectrometry assays demonstrating the direct binding of RvD2/Grp18 in muscle cells would strengthen this mechanistic aspect of the manuscript.

2- In vivo validation of molecular mechanism: To demonstrate that Grp18 is mediating the effects of RvD2 in vivo, it would be important to perform experiments showing that when Grp18 is deleted, the effects of the treatment are reduced/abrogated. While this reviewer acknowledges that crossing mdx mice with Grp18 knockout mice would require a long time, an experiments of acute injury in Grp18^{-/-} mice would address this issue. As it has been previously shown that RvD2 promotes tissue repair after acute injury, performing an acute injury in Grp18 knockout mice (and control wild-type mice) in the absence or presence of RvD2 treatment would demonstrate whether Grp18 is required for the observed beneficial effects on muscle repair, and strengthen the findings.

3- RvD2 levels in DMD: What are the levels of RvD2 in dystrophic mdx muscles? Is it lower than in healthy muscles?

4- Differentiation or fusion: The work shows that RvD2 treatment increases the number of differentiating muscle cells without affecting Pax7+ muscle stem cells in vivo. This would suggest that it acts mostly at the progenitor stage. What is the dynamic pattern of expression of Grp18 in muscle stem cells? Is it upregulated upon muscle injury and muscle stem cell activation? Does it persist in differentiated myogenin+ and myosin heavy chain+ cells? Does RvD2 treatment on

myotubes affect their size in vitro? These experiments would also help distinguish if the main effect of the treatment is on differentiation or during cell fusion.

5- Myofiber size: As the authors show an increased myofiber cross-section area upon RvD2 treatment in vitro and in vivo, it would be important to determine if this is the result of increased fusion of differentiating progenitors (increase in myonuclear number) or increase in myofiber size due to effects on the fiber itself. Inclusion of quantification of the number of myonuclei/fiber in histological sections would distinguish among these two possible mechanisms.

Specific points:

1- Figure 2B and 2G: a higher magnification of the immunofluorescence images for Pax7/myogenin and myosin heavy chain, or an inset with the magnification, would be useful to better visualize the results.

2- Figure 2D-E: the effect of RvD2 is less pronounced on muscle stem cells on isolated myofibers compared to isolated muscle stem cells (Figure 2B-C). Can the author discuss which contributing factors may reduce this effect when muscle stem cells are in their anatomical niche?

3- Figure 3B-D: the fusion index in control group is very low at ~15%. Is it due to initial low density of the cultures?

4- Figure 4B: a higher magnification of the immunofluorescence images or an inset with the magnification would be useful to better visualize the results.

5- Figure 5A: please provide the individual fluorescence channels for Pax7 and myogenin, not just the merge, or it is difficult to evaluate the images.

6- Figure 5B-C: please state if the total myogenic cells is the sum of Pax7+ and myogenin+ cells in the legend.

7- Figure 5A: the difference in eMyHC myofibers in the image is not striking, could you provide an image that more accurately reflects the quantification shown?

8- Figure 5A H&E: please provide better quality images for H&E staining (brighter), to more clearly visualize myofibers.

9- Figure S4: please provide representative immunofluorescence images of the IgG staining, to support the quantification shown.

10- Figure 6H-J: inclusion of prednisone treatment also in the experiment with mdx/utr^{-/-} mice would further strengthen the findings.

Reviewer #2:

Remarks to the Author:

In this study, Dort and colleagues analysed the effect of Resolvin-D2, a molecule that comes from the conversion of omega-3 fatty acids, in mitigating the different signs of Muscular Dystrophy (inflammation and muscle degeneration) in two dystrophic animal models, the mdx and the more severe mdx-utr^{-/-} mice.

The rationale is clear and the comparison with the effect of prednisone, the gold-standard treatment for DMD, was really appreciated.

This study has been approached by both in vitro and in vivo approaches and the development of the experiments logical.

However, this study is not particularly original, the results not that striking and the main message that arises from this work not clear in terms of results.

These are the main concerns

1. Does Resolvin-D2 mainly act on macrophages and, therefore, of the inflammatory process and, indirectly on myogenic cells, or directly on myogenic cells?

2. The authors demonstrated a direct effect of RvD2 on myogenic cells by increasing the amount of Myogenin+ cells and not on Pax7+ satellite cells. This might be dangerous, by unbalancing the proper ratio between satellite cell niche and myogenic cells. In a long-term period (not covered by this study) this might lead to an exhaustion of satellite cell pool and, therefore, to a no more beneficial effect.

3. The use of the more severe dystrophic animal model, the mdx-utrn dKO mice, was notable. However, the effect of RvD2 on mdx-utrn dKO mice was only described for the functional test with not that striking improvements (if slightly at the latest time points). The histological analysis and the RvD2 ability to mitigate the inflammation, as also its effect on macrophages, must be developed.

4. FIGURE 1.

The authors showed that RvD2 increased the proportion of macrophages (F4/80+) expressing the anti-inflammatory macrophage (MP) marker CD206. This characterization is too poor. This experiment should be addressed by polarizing the BM-derived macrophages to M1 and M2 and verifying the effect of RvD2 treatment in increasing the ratio between M1 and M2. This must be done even by a deeper analysis of M1/M2 markers. Additionally, to really prove the effect of RvD2 on MPs, a functional test should be performed by using conditioned medium of M1/M2 MPs treated or not with RvD2, on muscle cell proliferation and differentiation.

5. FIGURE SF1 and FIGURE 2G-H

It is not clear why RvD2 has not a direct effect on myoblast differentiation if not with the conditioned medium from treated-MP (SF1), whereas it does have in mdx myoblasts, without conditioned medium from MPs. This arises the main concern whether the RvD2 exerts its effect on MPs, and therefore on muscle cell, or directly on muscle cells.

6. A general comment: all the un vitro experiments should be performed even by comparing prednisone-treated myoblasts.

7. FIGURE 3H

the n is 2.

8. FIGURE 7

The model is too speculative since the mechanism proposed was entirely based on the in vitro experiments, which however are too preliminary and not well and deeply developed

Reviewer #3:

Remarks to the Author:

The study by Dort et al. investigates the effects of the resolvin-D2 on dystrophic muscle using two models of dystrophinopathy and assesses a direct effect of RvD2 on macrophages and muscle stem cells. The study is well carried out, is clearly presented, and the results are convincing. Because the use of resolvins is a promising avenue and presents several advantages as compared with classic anti-inflammatory drugs (e.g. glucocorticoids), additional experiments should complete the study:

- The expression of the gpr18 receptor should be evaluated in different cell types to see if RvD2 acts on other cell types than macrophages and MuSCs, for instance on FAPs and TRegs, which are very important in dmd muscle homeostasis. If these cell types express the receptor, the impact of RvD2 treatment on these cells should be evaluated as well.

- RvD2 does not affect MuSC expansion but stimulates their differentiation. Does the gpr18 expression varies with myogenesis (for instance no or low expression in myoblasts would explain no effect on their proliferation).

- Because the paper is written towards therapeutics, fibrosis should be analyzed in the severe model, or in diaphragm of mdx treated with RvD2. Also, a later time point than 21 days, (e.g. 2 or 3 months). Is the effect of RvD2 reversible?

Minor points

1) Introduction lines 61 to 63. "MuSC display globally impaired myogenesis capacity". This should be rephrased since no report of intrinsic defect of myogenesis has been reported in DMD cells yet, except for the self-renewal, in a previous study of the authors.

Ref 12: the paper by Blau in 1983 is made on non-purified cell preparation.

Ref 13: if not mistaken, this paper does not report a deficiency in DMD cells but suggests a role of CD82 in myogenesis.

Ref14: in this paper we may consider that the cells are in a specific environment

Ref15 does not compare mdx and WT (except for oxidative stress) in the myogenic capacity of MuSCs.

Therefore while there is clearly a defect in myogenesis in mdx, this is not demonstrated that this defect is intrinsic to the cells yet (except for the self-renewal).

2) Figure 1: the cartoon in A shows the isolation of macrophages from monocytes. But the text indicates BMDMs.

3) How the concentration of RvD2 was chosen for in vitro experiments? Is there a dose-dependent effect of RvD2? How the concentration for the in vivo experiments was chosen?

4) Sup Fig1 versus Figure 2H. SupFig1 shows no difference of myogenic cell fusion with medium containing RvD2 (unless it is degraded in the medium in 48h) while RvD2 has a stimulating effect of myogenin expression. What is the difference between the 2 culture set ups? This is not clear.

5) Figure1B is unreadable.

6) Figure 3. The fusion index is very different than that shown in Figure 2H. What does explain the difference ?

7) Please clarify the difference between Figure 6F and 6G. Is 6G the specific force? (so Y axis is wrong).

Point-by-point response

Reviewer #1:

“The findings are timely and novel, the experiments are rigorously performed, the manuscript is well written and the data clearly presented. However, additional experiments are required to fully support the authors’ interpretation.”

We would first like to thank the reviewer for his/her positive and constructive comments. We made significant addition to the manuscript according to the reviewer’s suggestions. This work led to the addition of 66 figure panels (for a total of 139 panels), including 2 additional main figures (for a total of 8 main figures), and 7 additional supplemental figures (for a total of 12 supplemental figures). All changes are highlighted in red in the manuscript. These changes strongly improved the manuscript and strengthen our conclusions.

1- Conclusive evidence of binding of RvD2 to Grp18 in muscle cells: While the authors show that acute downregulation or genetic deletion of Grp18 prevents the effect of RvD2 on mdx myoblasts on differentiation/fusion and Akt activation, a conclusive demonstration that it is mediated by direct binding in muscle cells is not provided. Inclusion of mass-spectrometry assays demonstrating the direct binding of RvD2/Grp18 in muscle cells would strengthen this mechanistic aspect of the manuscript.

The specific binding of RvD2 and Gpr18 in a native state (without detergent) using mass spectrometry approach as proposed is difficult to demonstrate in crude extract of myoblasts. However, the RvD2-GPR18 binding was thoroughly studied in immune cells, which highly express this receptor. First, the binding of the RvD2 to GPR18 in human leukocytes was demonstrated using recombinant GPR18 with tritiated RvD2 (Chiang *et al* 2015 J Exp Med, PMID: 26195725). Moreover, in this previous study, RvD2 receptors were screened using β -arrestin PathHunter GPCR system in the presence of 10 nM of RvD2 or vehicle control. The latter experiment screened over 70 GPR and GPR-like receptors. GPR18 was the receptor that clearly showed by far the strongest binding. RvD2 was also shown to stimulate intracellular cyclic AMP dependent on GPR18. Then, the functional binding of RvD2 was further confirmed in cells overexpressing GPR18. Specific binding of RvD2 to recombinant Gpr18 was confirmed using synthetic radiolabeled RvD2 in a competition binding assay. Moreover, it was shown that the effect of RvD2 is ablated in Gpr18-knockout mice (Chiang *et al.* 2017 J Immunol, PMID: 27994074). Other investigators have also shown that antagonist of GPR18 like O-1918 blocks the effects of RvD2 on different cell types *in vitro* such as neurons (Perna E *et al.* 2020, Gut, PMID: 33023902), keratinocytes (Hellmann J *et al.* 2018 J Invest Dermatol, PMID: 29559341), and endothelial cells (Zhang MJ *et al.* 2016 Circulation, PMID: 27507404). The ablation of the effect of RvD2 was also observed with the Gpr18 antagonist O-1918 in different *in vivo* models (Zuo G *et al.* 2018 Molecular Brain, PMID: 29439730; Deyama S *et al.* 2017 Int J Neuropsych, PMID: 28419244). Antagonist of other pro-resolving mediator receptors (e.g. antagonist of Fpr2, the receptor of Resolvin-D1) failed to block Resolvin-D2 effect (Perna E *et al.* 2020, Gut, PMID: 33023902).

Overall, the specificity of the binding RvD2/Gpr18 has been convincingly established previously from different groups using different methods, different models *in vitro* and *in vivo*, and different cell types. Our data strongly support that the mechanism is the same in our system. Our findings demonstrate that RvD2 effect is lost when Gpr18 is knocked down by siRNA, or in Gpr18-knockout cells. Moreover, in this revised version of the manuscript we also showed that the Gpr18 antagonist O-1918 blocks the therapeutic effect of RvD2 *in vivo* (see next point below). Altogether, findings from the literature published from different groups and using different methods combined with our experimental data using different approaches (knockdown, knockout, pharmacological inhibitor) strongly support that RvD2 mediates its effect through its interaction with Gpr18.

2- *In vivo* validation of molecular mechanism: To demonstrate that Gpr18 is mediating the effects of RvD2 *in vivo*, it would be important to perform experiments showing that when Gpr18 is deleted, the effects of the treatment are reduced/abrogated. While this reviewer acknowledges that crossing mdx mice with Gpr18 knockout mice would require a long time, an experiment of acute injury in Gpr18^{-/-} mice would address this issue. As it has been previously shown that RvD2 promotes tissue repair after acute injury, performing an acute injury in Gpr18 knockout mice (and control wild-type mice) in the absence or presence of RvD2 treatment would demonstrate whether Gpr18 is required for the observed beneficial effects on muscle repair, and strengthen the findings.

- We fully agree with the reviewer that the molecular mechanism of RvD2 *in vivo* is an important point to address. As mentioned by the reviewer, crossing Gpr18-knockout mice to mdx mice would have required a long time. We also believe that performing acute injury to Gpr18-knockout mice would not represent the complexity and the chronic degenerative environment observed in dystrophic muscles. Therefore, we opted for another approach to validate the molecular mechanism *in vivo*. We treated mdx mice with a highly specific Gpr18 antagonist (O-1918) 30 min before the administration of Resolvin-D2. This antagonist has been well-validated and is widely used in the field both *in vitro* and *in vivo* (Zuo G *et al.* 2018 *Molecular Brain*, PMID: 29439730; Deyama S *et al.* 2017 *Int J Neuropsychopharmacol*, PMID: 28419244; Offertaler L *et al.* 2003, *Mol Pharmacol*, PMID: 12606780; McHugh D *et al.* 2010 *BMC Neurosci*, PMID: 20346144). Our results confirm that the inhibition of Gpr18 with O-1918 ablated the therapeutic impact of Resolvin-D2 on muscle function of mdx mice (Figure 6K-M, attached below for ease of consultation).

Figure 6K-M: **K** Timeline of weekly *ip* injection of O-1918 (Gpr18 antagonist) prior to the injection of RvD2 (5 ug/kg) or vehicle. **L,M** Isometric contractile properties of EDL muscles showing the peak tetanic force (**L**) and specific force (**M**) of *mdx* mice treated weekly for 21 days. Data are presented as mean \pm SEM; $n=3$ mice per group; * $p < 0.05$.

3- RvD2 levels in DMD: What are the levels of RvD2 in dystrophic *mdx* muscles? Is it lower than in healthy muscles?

- This is a very interesting remark by the reviewer. Considering that healthy wildtype muscles and dystrophic *mdx* muscles are in different activation state (resting vs degenerating/regenerating), we did not only evaluate the expression of Resolvin-D2, but also other bioactive lipids by mass spectrometry to get a complete profile of pro-resolving vs pro-inflammatory lipids. Our findings show that the expression of Resolvin-D2 and other isoforms of Resolvins is similar between wildtype and *mdx* mice, but there is a strong increase in the expression of prostaglandins (PGE_2 , $PGF_{2\alpha}$) and other pro-inflammatory lipids (leukotriene- B_4 , thromboxane- B_2) in *mdx* muscles compared to WT (Figure 4A-C, and suppl. Fig. S6, attached below). These findings indicate that there is an imbalance in pro-inflammatory vs pro-resolving signals in dystrophic muscles, in favor of the former. This increase in prostaglandins/resolvin ratio has been observed in wildtype skeletal muscle in the early phase after an acute injury (at 1 day post-injury); however, in this healthy regenerating context it is quickly followed by a switch toward pro-resolving mediators leading to a reduction in the prostaglandins/resolvins ratio (as soon as day 2 after the injury) (Giannakis N *et al.* 2019 Nat Immunol: PMID: 30936495). These findings suggest that dystrophic muscles are stuck in a pro-inflammatory stage. The lack of pro-resolving signals in *mdx* muscles supports the idea that Resolvin-D2 administration could help to restore the resolution of inflammation that is normally observed in healthy regenerating muscles. In support of this hypothesis, our latest results indicate that the administration of RvD2 to macrophages *in vitro* decreases their expression of COX-2 (*Ptgs2* gene) and prostaglandin-E synthase (*Ptges* gene), the two main enzymes responsible of PGE_2 expression (Fig. 1K, attached below). Therefore, the administration of RvD2 could reduce the prostaglandins/resolvins ratio in *mdx* muscles by increasing the concentration of resolvins *per se*, and also by decreasing the endogenous production of prostaglandins *in situ*.

Figure 4A-C: Quantification by mass spectrometry (LC-MS/MS) of **A**) Resolvin-D isoforms (RvD1, RvD2, RvD3, 17(R,S)-RvD4, RvD5), and **B**) prostaglandins isoforms (PGE₂, PGF_{2α}) in the gastrocnemius muscle of wildtype (WT) and mdx mice. **C**) Ratio of prostaglandins/Resolvins in TA muscle of wildtype and mdx mice. Data are presented as mean ± SEM; n=3 mice per group; **p* < 0.05.

Supplemental Figure S6: Expression of bioactive lipids in dystrophic muscles. Mass spectrometry (LC-MS/MS) analysis of pro-inflammatory and pro-resolving bioactive lipids expression in the *tibialis anterior* muscle of mdx and wildtype mice. Pro-inflammatory mediators: TXB₂ (thromboxane-B₂), LTB₄ (Leukotriene-B₄), PGE₂ (Prostaglandin-E₂), PGF_{2α} (Prostaglandin-F_{2α}). Pro-resolving mediators: Resolvin-D1 (RvD1), -D2, -D3, 17(R,S)-RvD4, -D5, Resolvin-E1 (RvE1). Data are presented as mean ± SEM; n=3 biologically independent samples; **p*<0.05 compared to wildtype.

Fig 1K: Gene expression of the pro-inflammatory markers *Ptgs2*, *Gpr18*, *Ptges*, and *Cd80* on M1-polarized macrophages treated with or without RvD2 for 48 h. Data are presented as mean ± SEM; n=3 mice per group; **p* < 0.05.

4- Differentiation or fusion: The work shows that RvD2 treatment increases the number of differentiating muscle cells without affecting Pax7+ muscle stem cells *in vivo*. This would suggest that it acts mostly at the progenitor stage. What is the dynamic pattern of expression of Grp18 in muscle stem cells? Is it upregulated upon muscle injury and muscle stem cell activation? Does it persist in differentiated myogenin+ and myosin heavy chain+ cells? Does RvD2 treatment on myotubes affect their size *in vitro*? These experiments would also help distinguish if the main effect of the treatment is on differentiation or during cell fusion.

- We measured Gpr18 expression at different stages of myogenesis as suggested by the reviewer. Our findings from western blot indicated that Gpr18 expression is increased during the switch from proliferative myoblasts to differentiated myoblasts (Figure 3A,B, attached below). Immunostaining on skeletal muscle sections also confirmed that Gpr18 is expressed in Myog⁺ cells *in vivo* (Suppl. Fig. S5B,D, attached below). These findings are consistent with our results showing that Resolvin-D2 has no effect on myoblast proliferation, but rather stimulate myoblast differentiation (Fig. 2A-I). Our findings also indicate that RvD2 acts on myoblast cell fusion by increasing myosin heavy chain content and fusion index (Fig 2J-M) as well as myotube size (Fig 3F). To determine if RvD2 act directly on myotubes, we treated myotubes (already formed for 2 days in differentiation medium) for 48h with RvD2. We observed that RvD2 induced an increase in myotube size, albeit at a lower level than when RvD2 is added at the start of the differentiation process (Suppl. Fig. S4, attached below). Altogether, these results indicate that the main effects of Resolvin-D2 is mediated at the progenitor stage, as proposed by the reviewer. We modified the text in the manuscript to reflect this fact more accurately.

Figure 3A,B: **A)** Representative western blot and **B)** quantification of Gpr18 expression in proliferating myoblasts and differentiated myoblasts (1 day in differentiating medium) isolated from mdx mice. Data are presented as mean \pm SEM; n=5 biologically independent samples; **p<0.01.

Suppl. Fig S5B,D: **B)** Representative pictures of Myog (differentiated myoblast marker; red), Gpr18 (green), and DAPI (blue). Arrowhead shows double positive cells (Myog⁺ Gpr18⁺). **D)** Representative pictures of Myog (orange), CD25 (Treg marker, red), Gpr18 (green), and DAPI, showing that Myog⁺ myogenic cells express Gpr18 (arrowheads) but not CD25⁺ Treg (arrows).

Supplemental Figure S4: Effect of RvD2 on myotubes. Myotubes were differentiated for 2 days. Thereafter, they were treated with RvD2 or vehicle (Ctrl) for an additional 2 days, and myotube diameter was measured. Data are presented as mean \pm SEM; n=3 biologically independent samples performed in technical duplicates; ***p<0.001, compared to vehicle.

5- Myofiber size: As the authors show an increased myofiber cross-section area upon RvD2 treatment in vitro and in vivo, it would be important to determine if this is the result of increased fusion of differentiating progenitors (increase in myonuclear number) or increase in myofiber size due to effects on the fiber itself. Inclusion of quantification of the number of myonuclei/fiber in histological sections would distinguish among these two possible mechanisms.

- We performed the quantification of myonuclei/fiber as suggested by the reviewer. Our results show that Resolvin-D2 increases the proportion of nuclei per fiber compared to vehicle or prednisone-treated *mdx* mice after 21 days of treatment (Fig. 5H). We also performed long-term experiments, and we have observed that the therapeutic effects of RvD2 are maintained after 2 months of treatment, including the higher number of myonuclei per fiber (Fig. 7E). These results suggest that myonuclear accretion contributes to the increase in myofiber size.

Figure 5H: Number of myonuclei per fiber after 21 days of daily intraperitoneal (*ip*) injection of Resolvin-D2 (RvD2, 5 ug/kg/day), prednisone (2 mg/kg/day) or vehicle (Ctrl) in *mdx* mice. Data are presented as mean \pm SEM; n=4 mice per group; *p<0.05.

Figure 7E: Number of myonuclei per fiber after 2 months of weekly intraperitoneal (*ip*) injection of Resolvin-D2 (RvD2, 5 ug/kg), prednisone (2 mg/kg) or vehicle (Ctrl) in *mdx* mice. Data are presented as mean \pm SEM; n=4 mice per group; *p<0.05 **p<0.01.

Specific points:

1- Figure 2B and 2G: a higher magnification of the immunofluorescence images for Pax7/myogenin and myosin heavy chain, or and inset with the magnification, would be useful to better visualize the results.

- Higher magnification images were provided as requested (now Fig 2B,J)

2- Figure 2D-E: the effect of RvD2 is less pronounced on muscle stem cells on isolated myofibers compared to isolated muscle stem cells (Figure 2B-C). Can the author discuss which contributing factors may reduce this effect when muscle stem cells are in their anatomical niche?

- Analysis of Gpr18 indicates that the expression of this receptor is upregulated during later stages of myogenesis. Therefore, the stronger impact of RvD2 on primary myoblasts vs isolated muscle fibers likely reflect the fact that primary myoblasts are at a more advanced stage of myogenesis (they have been actively cycling *in vitro*) compared to isolated myofibers in which quiescent muscle stem cells are entering cell cycle. This difference has been addressed in the discussion of the manuscript: "The effect of RvD2

on myoblast differentiation is less pronounced in ex vivo single muscle fibers than in primary myoblasts in vitro, which suggest that RvD2 preferentially targets myogenic cells at the progenitor stage rather than quiescent/activating MuSC. The superior impact of RvD2 on myogenic progenitors is consistent with the increased expression of Gpr18 during myoblast differentiation. These findings also explain that RvD2 does not promote the exhaustion of the MuSC pool in vivo, even at long-term (2 months of treatment)."

3- Figure 3B-D: the fusion index in control group is very low at ~15%. Is it due to initial low density of the cultures?

- In figure 3, the cells were transfected (siRNA) prior to differentiation, which might have stressed the cells and delayed fusion. However, we did not notice a reduction in cell density in the different groups. More importantly, considering that all the groups were treated equally (all groups were transfected with siRNA scrambled or siGpr18), we are convinced that the results are valid.

4- Figure 4B: a higher magnification of the immunofluorescence images or an inset with the magnification would be useful to better visualize the results.

- Higher magnification images were provided as requested (now Fig. 4E)

5- Figure 5A: please provide the individual fluorescence channels for Pax7 and myogenin, not just the merge, or it is difficult to evaluate the images.

- Pax7 and Myog staining were provided in individual channels as requested (now Fig 5B)

6- Figure 5B-C: please state if the total myogenic cells is the sum of Pax7+ and myogenin+ cells in the legend.

- The figure legend was modified according to the reviewer's suggestion (now Suppl. Fig. S10).

7- Figure 5A: the difference in eMyHC myofibers in the image is not striking, could you provide an image that more accurately reflects the quantification shown?

- We provided higher magnification images that are representative of the proportion of eMyHC⁺ fibers (now Fig 5B).

8- Figure 5A H&E: please provide better quality images for H&E staining (brighter), to more clearly visualize myofibers.

- Brighter images with better contrast were provided to allow the clear visualization of myofibers (now Fig. 5B).

9- Figure S4: please provide representative immunofluorescence images of the IgG staining, to support the quantification shown.

- We included representative images of IgG staining as requested by the reviewer (now Suppl. Fig. S11)

10- Figure 6H-J: inclusion of prednisone treatment also in the experiment with *mdx/utr*^{-/-} mice would further strengthen the findings.

- We included a prednisone treatment in *mdx-utr* dKO mice as suggested (now Fig. 8, attached below). Furthermore, we performed immunohistological assessments (macrophages count, myogenic cell count) of the treated mice in addition to the functional assessments (*in vivo* and *ex vivo* force measurements). These results confirm our findings observed in *mdx* mice, i.e. that Resolvin-D2 dampen inflammation to a similar level than prednisone (Fig. 8B-D), but it increases the generation of differentiated myoblasts compared to prednisone (Fig. 8B,E,F). These changes results in higher muscle function in RvD2-treated mice compared to prednisone-treated mice or control mice (Fig. 8G,H).

Figure 8. The superior therapeutic effect of RvD2 is maintained in the severely affected *mdx-utr* dKO mouse model. **A)** Timeline of weekly *ip* injection of either RvD2 (5 ug/kg), prednisone (2 mg/kg) or vehicle into *mdx-utr* dKO mice for 3 weeks. **B)** Representative images of F4/80 (red) (upper panel) and CD206 (green) (middle panel), or Pax7 (red) and Myog (green) (lower panel) co-immunostaining in muscle section of *tibialis anterior* muscle of treated *mdx-utr* dKO mice. Scale bars = 100 μ m. **C,D)** Quantification of total number of F4/80⁺ macrophages (**C**) and proportion of anti-inflammatory macrophages (F4/80 CD206⁺) (**D**) in TA muscles of *mdx-utr* dKO mice. **E,F)** Quantification of total number of Pax7⁺ MuSC (**E**) and Myog⁺ differentiated myoblasts (**F**) in TA muscles of *mdx-utr* dKO mice. **G)** Hang test performance (time mice can hold on an inverted grid) of *mdx-utr* dKO mice treated for 3 weeks. **H)** Specific muscle force (N/cm²) of EDL muscles of *mdx-utr* dKO mice treated for 3 weeks. Data are presented as mean \pm SEM; n=3-4 mice per group; **C-G)** *p < 0.05; **P < 0.01. **H)** *p < 0.05 compared with vehicle; † p < 0.05 compared with prednisone.

Reviewer #2 (Remarks to the Author):

“The rationale is clear and the comparison with the effect of prednisone, the gold-standard treatment for DMD, was really appreciated (...) However, this study is not particularly original, the results not that striking and the main message that arises from this work not clear in terms of results.”

We thank the reviewer for his/her constructive comments about our work. We made significant changes according to the reviewer's comments. This work led to the addition of 66 figure panels (for a total of 139 panels), including 2 additional main figures (for a total of 8 main figures), and 7 additional supplemental figures (for a total of 12 supplemental figures). All changes are highlighted in red in the manuscript. We believe that this revised version of the manuscript is now strongly improved, and that the main message is clearer.

1. Does Resolvin-D2 mainly act on macrophages and, therefore, of the inflammatory process and, indirectly on myogenic cells, or directly on myogenic cells?

- Our results *in vitro* suggest that RvD2 acts on myogenic cells through both direct and indirect mechanisms. *In vitro* experiment indicate that paracrine factors secreted by RvD2-treated macrophages stimulate myogenesis (Fig 1). Moreover, treatment of myogenic cells *in vitro* also directly promotes myogenic cell differentiation and fusion (Fig 2). However, analysis of *in vivo* experiments indicates that the direct mechanism of RvD2 on myogenic cells is mainly responsible of its higher therapeutic capacity compared to prednisone. Indeed, prednisone and RvD2 have very similar impact on inflammation (similar reduction in macrophages/neutrophils/lymphocytes density, and switch in macrophage phenotype) (Fig. 4D-M). The effect of RvD2 on fibrosis and necrosis was also similar to the one of prednisone (Fig. 5B,M,N and Suppl. Fig. S11). The only notable difference between RvD2-treated mice and prednisone-treated mice that could explain the higher functional outcomes, is that RvD2 increases the myogenic cell pool and their fusion to regenerating myofibers (Fig. 5A-L and Suppl. Fig. S10). We clarified this rationale in the discussion: “*Considering that RvD2 and glucocorticoids have similar anti-inflammatory and anti-fibrotic effects on dystrophic muscles, our findings suggest that the superior therapeutic effect of RvD2 is largely attributable to its capacity to directly target myogenic cells in order to improve myogenesis.*”

2. The authors demonstrated a direct effect of RvD2 on myogenic cells by increasing the amount of Myogenin+ cells and not on Pax7+ satellite cells. This might be dangerous, by unbalancing the proper ratio between satellite cell niche and myogenic cells. In a long-term period (not covered by this study) this might lead to an exhaustion of satellite cell pool and, therefore, to a no more beneficial effect.

- We fully understand the reviewer's concern. Therefore, we added an extra time point after 2 months of treatment, to determine the impact of RvD2 at long term (Fig. 7, attached below for ease of consultation). Our results showed that even at this longer time point, we did not observe any reduction in the muscle stem cell pool (Pax7⁺ cells) in RvD2 treated mice (Fig. 7B,C). Actually, the number of Pax7⁺ cells was higher in RvD2 treated mice compared to prednisone-treated mice (Fig. 7B,C), which is consistent with the negative impact of prednisone on myogenic cell proliferation that we observed *in vitro* (Fig. 2A,E-G). Similar to what we observed at 1 and 3 weeks in treated *mdx* mice, the number of differentiated myoblasts (Myog⁺ cells) was still increased by about 3-fold after 2 months of RvD2 treatment (Fig. 7D). Moreover, the increase in muscle force *in vivo* and *ex vivo* was still significantly higher in RvD2-treated mice than in control or prednisone-treated mice (Fig. 7G,H).

It was not possible to perform longer time points due to time constraint (especially in time of Covid-19, where our mouse colonies have been reduced and there was a constant risk of shutdown). However, these results show that there is no signs of exhaustion of the MuSC pool or reduced therapeutic efficacy after 2 months of treatment, which suggests that the beneficial effects could be maintained for even longer period. These results are consistent with our recent observation that Gpr18 expression is increased during myoblast differentiation (Fig. 3A,B, attached below), which indicates that RvD2 targets preferentially myogenic cells at the progenitor stage, thereby preventing the exhaustion of the MuSC pool.

Figure 7. The superior therapeutic effect of RvD2 is maintained at long-term in *mdx* mice. **A)** Timeline of weekly *ip* injection of either RvD2 (5 ug/kg), prednisone (2 mg/kg) or vehicle into *mdx* mice for 2 months. **B)** Representative images of Pax7, Myog, Laminin, Collagen (Sirius Red) immunostaining in muscle section of *tibialis anterior* muscle (for Pax7, Myog, Laminin) or diaphragm muscle (for collagen staining) of treated *mdx* mice. Scale

bars = 100 μm . **C,D**) Quantification of total number of Pax7⁺ MuSC (**C**) and Myog⁺ differentiated myoblasts (**D**) in TA muscles of *mdx* mice treated for 2 months. **E**) Number of myonuclei per fiber in TA muscles of *mdx* mice treated for 2 months. **F**) Quantification of Sirius Red positive area (collagen) in diaphragm muscles of *mdx* mice treated for 2 months. **G**) Grip strength measured in treated *mdx* mice after 7 weeks of treatment. **H**) Specific muscle force (N/cm²) of EDL muscles of *mdx* mice treated for 2 months. Data are presented as mean \pm SEM; n=4 mice per group; **C-G**) *p < 0.05; **p < 0.01. **H**) *p < 0.05 compared with vehicle; † p < 0.05 compared with prednisone.

Figure 3A,B: **A**) Representative western blot and **B**) quantification of Gpr18 expression in proliferating myoblasts and differentiated myoblasts (1 day in differentiating medium) isolated from *mdx* mice. Data are presented as mean \pm SEM; n=5 biologically independent samples; **p<0.01.

3. The use of the more severe dystrophic animal model, the *mdx-utrn* dKO mice, was notable. However, the effect of RvD2 on *mdx-utrn* dKO mice was only described for the functional test with not that striking improvements (if slightly at the latest time points). The histological analysis and the RvD2 ability to mitigate the inflammation, as also its effect on macrophages, must be developed.

- We performed immunohistological measurements on *mdx-utrn* dKO mice as suggested by the reviewer. The results are very similar to what we observed in *mdx* mice (i.e. reduction in macrophage density, increase in M2 macrophage proportion, increase in Myog⁺ cell density) (Fig. 8, attached below). We also assessed *ex vivo* muscle force (which is more specific and responsive to change in muscle force than the hang test) that shows a more striking improvement in RvD2-treated mice.

Figure 8. The superior therapeutic effect of RvD2 is maintained in the severely affected *mdx-utrn* dKO mouse model. **A)** Timeline of weekly *ip* injection of either RvD2 (5 ug/kg), prednisone (2 mg/kg) or vehicle into *mdx-utrn* dKO mice for 3 weeks. **B)** Representative images of F4/80 (red) (upper panel) and CD206 (green) (middle panel), or Pax7 (red) and Myog (green) (lower panel) co-immunostaining in muscle section of *tibialis anterior* muscle of treated *mdx-utrn* dKO mice. Scale bars = 100 μ m. **C,D)** Quantification of total number of F4/80⁺ macrophages (**C**) and proportion of anti-inflammatory macrophages (F4/80 CD206⁺) (**D**) in TA muscles of *mdx-utrn* dKO mice. **E,F)** Quantification of total number of Pax7⁺ MuSC (**E**) and Myog⁺ differentiated myoblasts (**F**) in TA muscles of *mdx-utrn* dKO mice. **G)** Hang test performance (time mice can hold on an inverted grid) of *mdx-utrn* dKO mice treated for 3 weeks. **H)** Specific muscle force (N/cm²) of EDL muscles of *mdx-utrn* dKO mice treated for 3 weeks. Data are presented as mean \pm SEM; n=3-4 mice per group; **C-G)** *p < 0.05; **p < 0.01. **H)** *p < 0.05 compared with vehicle; † p < 0.05 compared with prednisone.

4. FIGURE1.

The authors showed that RvD2 increased the proportion of macrophages (F4/80+) expressing the anti-inflammatory macrophage (MP) marker CD206. This characterization is too poor. This experiment should be addressed by polarizing the BM-derived macrophages to M1 and M2 and verifying the effect of RvD2 treatment in increasing the ratio between M1 and M2. This must be done even by a deeper analysis of M1/M2 markers. Additionally, to really prove the effect of RvD2 on MPs, a functional test should be performed by using conditioned medium of M1/M2 MPs treated or not with RvD2, on muscle cell proliferation and differentiation.

- We agree with the reviewer that macrophages are a complex and heterogenous population that needs to be carefully characterized. We added a full set of experiments on RvD2-treated macrophages as suggested by the reviewer (new Fig 1, attached below). First, we polarized macrophages in M1 macrophages using LPS + IFN- γ activation, which has been widely described in the field (Mosser DM & Zhang X 2008, *Curr Protoc Immunol*, PMID: 19016446; Jablonski KA *et al* PLoS One 2015, PMID: 26699615). We did not

polarize macrophage toward the M2 phenotype (for instance using IL4 or IL13 administration) because Gpr18 (RvD2 receptor) is almost exclusively expressed in M1 macrophages (one of the most enriched gene in M1 compared to M2 macrophages (Jablonski KA *et al* PLoS One 2015, PMID: 26699615; Orecchioni M *et al* 2019 Front Immunol, PMID: 31178859). Moreover, we would not expect RvD2 to have an important impact on M2-polarized macrophages as it cannot induce a switch in macrophage phenotype if these macrophages are already strongly polarized toward M2 phenotype. Thus, we treated M1-polarized macrophages with RvD2 and carefully assessed the changes in macrophage phenotype using different methods. First, in addition to CD206 we also performed immunostaining for two other well-known M2 macrophage markers, Arginase1 (Arg1) and CD163 (Deng B *et al*, 2012 J Immunol, PMID: 22933625; Mounier R *et al*, 2013 Cell Metab, PMID: 23931756). The expression of all three markers was increased in RvD2 treated macrophages (Fig. 1B-F). Moreover, we assessed the expression of iNOS, a well characterized M1 macrophage marker, by flow cytometry (Villalta SA *et al*, 2009 Hum Mol Genet, PMID: 18996917). Our results showed a strong reduction of iNOS expression in RvD2 treated macrophages (Fig. 1G-I). Furthermore, we assessed the expression of multiple genes identified by transcriptomics analysis to be specifically expressed in the *in vivo* and *in vitro* signatures of M1 or M2 macrophages (Jablonski KA *et al* PLoS One 2015, PMID: 26699615). Our results showed a downregulation in M1 gene signature (reduction in the expression of *Cd80*, *Gpr18*, *Ptgs2*, and *Ptges*) and an increase in M2 gene signature (increase in the expression of *Cd163*, *Pparg*, *Chil3*, and *Anxa1*) (Fig. 1J,K). The changes in gene expression are similar between RvD2 and prednisone (Suppl. Fig. S1, attached below), further supporting the idea that both molecules have a similar effect on inflammation (and that the superior therapeutic effect of RvD2 is due to its direct impact on myogenic cells). The impact of RvD2-treated M1-polarized macrophages culture medium on myoblast proliferation and differentiation/fusion was assessed as suggested by the reviewers (Fig. 1L-O). RvD2-conditioned medium from M1-polarized macrophages did not affect cell proliferation, but promoted myoblast differentiation and fusion compared to untreated M1-polarized macrophages (similar to what we previously observed in unpolarized macrophages; Suppl. Fig. S2).

Figure 1. Resolvin-D2 promotes the switch in macrophage phenotype and their release of pro-myogenic factors. **A**) Graphic overview of the myoblast:macrophage-conditioned medium co-culture experiment (image created with BioRender.com). Monocytes purified from bone marrow of *mdx* mice are differentiated into macrophages (M-CSF), polarized into M1 macrophages (IFN- γ and LPS supplementation) for 24h and treated with Resolvin-D2 (RvD2; 200 nM) for 48h. **B**) Representative images of immunofluorescence performed on macrophages *in vitro* for F4/80 (pan-macrophage marker; red), and the anti-inflammatory markers CD206 (green), and Arginase-1 (Arg1; yellow), and **C**) for F4/80 (green) and CD163 (red), and DAPI (blue). Scale bars = 50 μ m. **D-F**) Percentage of total macrophages (F4/80⁺ cells) co-expressing the anti-inflammatory macrophages markers CD206 (**D**), or Arg1 (**E**), or CD163 (**F**) following 48 h treatment with RvD2 or control. **G,H**) Representative FACS plot showing the expression of the pan-macrophage marker F4/80 (FITC, y-axis) and the pro-inflammatory marker iNOS (PE, x-axis) on M1-polarized macrophages treated with RvD2 or vehicle. **I**) Quantification of the proportion of F4/80⁺ iNOS⁺ macrophages. **J,K**) Gene expression of **J**) the anti-inflammatory markers *Anxa1*, *Cd163*, *Pparg*, and *Chil3* and **K**) the pro-inflammatory markers *Ptgs2*, *Gpr18*, *Ptges*, and *Cd80* on M1-polarized macrophages treated with or without RvD2 for 48 h. **L**) Representative images and **M**) quantification of Ki67 immunostaining performed on primary myoblasts incubated with conditioned medium from macrophages treated with RvD2 or vehicle. Scale bars = 50 μ m. **N**) Representative images of myoblasts differentiated into myotubes for 4 days with the

macrophage-conditioned medium and stained for myosin heavy chain (MyHC, green) and DAPI (blue). Scale bars = 50 μm . **O**) Quantification of the fusion index (proportion of nuclei into multinucleated myotubes / total nuclei). Data are presented as mean \pm SEM; $n=3$ biologically independent samples performed in technical duplicates; * $p<0.05$, ** $p<0.01$.

Supplemental Figure S1. Impact of prednisone on the phenotype of M1-polarized macrophages. Monocytes purified from bone marrow of *mdx* mice are differentiated into macrophages (M-CSF), polarized into M1 macrophages (IFN- γ and LPS supplementation) and treated with prednisone (pred; 10 μM) for 48h. **A**) Gene expression of the pro-inflammatory markers *Ptgs2*, *Gpr18*, *Ptges*, and *Cd80*, and **B**) gene expression of the anti-inflammatory markers *Anxa1*, *Cd163*, *Pparg*, and *Chil3* on M1-polarized macrophages treated with or without prednisone for 48 h. Data are presented as mean \pm SEM; $n=3$ biologically independent samples performed in technical duplicates; * $p<0.05$, ** $p<0.01$, *** $p<0.001$

5. FIGURE SF1 and FIGURE 2G-H

It is not clear why RvD2 has not a direct effect on myoblast differentiation if not with the conditioned medium from treated-MP (SF1), whereas it does have in *mdx* myoblasts, without conditioned medium from MPs. This arises the main concern whether the RvD2 exerts its effect on MPs, and therefore on muscle cell, or directly on muscle cells.

- Previous studies have shown that resolvins have a relatively short half-life of a few hours (Krashia P *et al*, 2019 Nat Comm, PMID: 31477726; Giannakis N *et al* 2019 Nat Immunol: PMID: 30936495). Therefore, RvD2 added in the macrophage medium has likely been degraded over the 48h incubation. To confirm the degradation of RvD2, we analyzed by ELISA the concentration of RvD2 after 48h of incubation. We observed that the concentration of RvD2 at 48h is approximately 10% of the initial concentration (200nM)(Suppl. Fig. S3A, attached below). Therefore, the remaining levels of RvD2 in macrophage conditioned medium are too low to induce a significant impact directly on myogenic cells (Suppl. Fig. S3B).

Supplemental Figure S3. Myogenic effects of macrophage-conditioned medium are not mediated by remaining Resolvin-D2 in the medium. **A**) Concentration of Resolvin-D2 (RvD2) was assessed by ELISA in the culture medium 48h after the supplementation of RvD2. The final concentration was expressed as percentage on initial concentration

(200nM). **B)** To determine if the low remaining level of RvD2 in the medium after 48 h of treatment affects myogenesis, we added RvD2 in the medium in a macrophage-free well for 48 h. The medium was added to differentiating myoblasts. Data showed no difference compared to control, indicating that the myogenic effect of the macrophage-conditioned media is driven by paracrine factors secreted by macrophages upon RvD2 treatment and not by remaining RvD2 in the medium. Data are presented as mean \pm SEM; n=3-4 biologically independent samples.

6. A general comment: all the *in vitro* experiments should be performed even by comparing prednisone-treated myoblasts.

- This is another important point raised by the reviewer. We assessed the impact of prednisone on myoblasts *in vitro* as suggested by the reviewer (Fig 2, attached below). The results showed that prednisone has a negative impact on myoblast proliferation as shown with other glucocorticoids by other groups (te Pas MF *et al*, 2000 Mol Biol Rep, PMID: 11092555; Dong Y *et al*, PLoS One 2013, PMID: 23516508). These data strengthen our conclusion that RvD2 has a superior impact on myogenesis compared to prednisone.

Figure 2. Resolvin-D2 directly targets myogenic cells and stimulates myogenesis. **A)** Primary myoblasts isolated from *mdx* mice were plated *in vitro* in an automated live imaging analysis system (IncuCyte). Cells were cultured in proliferation media supplemented with Resolvin-D2 (RvD2, 200 nM), prednisone (10 μ M) or vehicle, and cell confluence was assessed for 3 days. **B-D)** Dystrophin-deficient myoblasts were cultured in low serum medium for 16 h supplemented with RvD2, prednisone, or vehicle. **B)** Representative images of immunofluorescence for Pax7 (red) and Myog (green). Scale bars = 50 μ m. **C,D)** Quantification of the proportion of Pax7-expressing cells (MuSC/proliferative myoblasts) and Myog-expressing cells (differentiated myoblasts). **E-I)** Single myofibers were isolated from the EDL muscle of *mdx* mice and cultured for 72 h with RvD2 (200 nM), prednisone (10 μ M) or vehicle (n=30 fibers/biological samples). **E)** Representative images of myofibers immunostained for Pax7 (red) and Myog (green). Scale bars = 50 μ m. **F,G)** Quantification of the total number of Pax7⁺ cells and Myog⁺ cells per fiber. **H,I)** Proportion of pax7-expressing and myogenin-expressing cells relative to the total number of myogenic cells (Pax7⁺ and Myog⁺). **J,K)** Dystrophin-deficient myoblasts were differentiated for 4 days in low serum medium supplemented

with RvD2, prednisone, or vehicle. **J**) Representative images of myotubes immunostained for MyHC (green) and DAPI (blue). Scale bars = 100 μm . **K**) Quantification of the fusion index (proportion of nuclei into multinucleated myotubes / total nuclei). **L,M**) Representative images and quantification of Myosin heavy chain expression by Western Blot in myotubes (relative to β -actin as loading control). Data are presented as mean \pm SEM; $n=3$ biologically independent samples performed in technical duplicates; * $p<0.05$, ** $p<0.01$.

7. FIGURE 3H

the n is 2.

- We apologize for the $n=2$. We had not been able to complete this experiment in the initial version of the manuscript because our research center was closed during that period due to the Covid-19 pandemic. We completed this experiment as soon as we came back at the bench. Additional data confirmed that the Akt pathway is not activated by RvD2 in Gpr18-KO myoblasts (Fig 3I,J).

8. FIGURE 7

The model is too speculative since the mechanism proposed was entirely based on the *in vitro* experiments, which however are too preliminary and not well and deeply developed

- In this revised version, we have expanded the *in vitro* experiments and we added several key experiments to confirm the mechanism of RvD2 *in vivo*. *In vitro*, we deeply characterized the impact of RvD2 on M1-polarized macrophages (Fig 1), and we showed that while RvD2 stimulate myogenesis *in vitro*, prednisone reduces myogenic cell proliferation (Fig 2). *In vivo*, we showed that RvD2 and prednisone have similar effect on inflammation, necrosis, and fibrosis; and that the superior therapeutic potential of RvD2 is mediated by their stimulating effect on myogenic cells (Figs 4 and 5). Moreover, we demonstrated that the pretreatment of *mdx* mice with the Gpr18 antagonist O-1918 ablates the effect of RvD2 *in vivo* (Fig. 6K-M, attached below). Therefore, we believe that the mechanism proposed is strongly strengthen in this revised version of the manuscript. However, considering that in this revised version of the manuscript we added numerous main and supplemental figures, as well as dozens of additional figure panels we decided to remove this schematic from the manuscript for the sake of clarity.

Figure 6K-M: **K**) Timeline of weekly *ip* injection of O-1918 (Gpr18 antagonist) prior to the injection of RvD2 (5

ug/kg) or vehicle. **L,M**) Isometric contractile properties of EDL muscles showing the peak tetanic force (**L**) and specific force (**M**) of *mdx* mice treated weekly for 21 days. Data are presented as mean \pm SEM; n=3 mice per group; * $p < 0.05$.

Reviewer #3 (Remarks to the Author):

“The study is well carried out, is clearly presented, and the results are convincing. Because the use of resolvins is a promising avenue and presents several advantages as compared with classic anti-inflammatory drugs (e.g. glucocorticoids), additional experiments should complete the study.”

We would first like to thank the reviewer for his/her positive and constructive comments. We made significant addition to the manuscript according to the reviewer's suggestions. This work led to the addition of 66 figure panels (for a total of 139 panels), including 2 additional main figures (for a total of 8 main figures), and 7 additional supplemental figures (for a total of 12 supplemental figures). All changes are highlighted in red in the manuscript. These changes strongly improved the manuscript and strengthen our conclusions.

- The expression of the gpr18 receptor should be evaluated in different cell types to see if RvD2 acts on other cell types than macrophages and MuSCs, for instance on FAPs and TRegs, which are very important in dmd muscle homeostasis. If these cell types express the receptor, the impact of RvD2 treatment on these cells should be evaluated as well.

- This is an important point raised by the reviewer. To evaluate which cell types express Gpr18 in skeletal muscle we performed a co-immunostaining for Gpr18 and different cellular markers (Suppl. Fig. S5, attached below for ease of consultation). Immunostaining confirmed that Gpr18 is expressed by a large subset of macrophages (Suppl. Fig. S5A). Considering that Gpr18 is almost exclusively expressed in M1 macrophages (Jablonski KA *et al* PLoS One 2015, PMID: 26699615; Orecchioni M *et al* 2019 Front Immunol, PMID: 31178859), these F4/80⁺Gpr18⁺ likely represents pro-inflammatory macrophages. We also noticed that Gpr18 is expressed in myogenic cells, preferentially in Myog⁺ myoblasts (Suppl. Fig. S5B,D). Analysis from Western blot confirmed that Gpr18 expression is increased in differentiating myoblasts compared to proliferating myoblasts (Fig. 3A,B; see next point below). We also assessed Gpr18 expression in CD25⁺ Treg and found virtually no positive cells (Suppl. Fig. S5C,D). This result is coherent with a paper showing by transcriptomics analysis that Gpr18 is one of the most downregulated gene in Treg isolated from skeletal muscles compared to Treg from spleen (Burzyn D *et al*, 2013 Cell, PMID: 24315098). Accordingly, we assessed the density of Treg by immunofluorescence (CD3⁺Foxp3⁺) in treated muscles and found that RvD2 did not significantly affect the number of Tregs (Suppl. Fig. S7).

We also looked at the expression of Gpr18 in FAPs by western blot. Our results showed that Gpr18 is expressed in FAPs but at much lower level (8-fold decrease) than in proliferating myoblasts (in which the expression of Gpr18 is already low compared to differentiated myoblasts; see next point) (Suppl. Fig. S5E,F). We further assessed the effect of RvD2 administration on FAPs proliferation *in vitro* and we did not observe any difference (Suppl. Fig S5G). Considering that FAPs are the main source of fibrosis in *mdx*

mice (Ieronimakis N *et al*, 2016 J Pathol, PMID: 27569721), we also assessed fibrosis *in vivo*. We observed that RvD2 induced a decrease in fibrosis that is similar to the one of glucocorticoids (Fig. 5 B,M,N). These findings indicate that the effect of RvD2 on FAPs/fibrosis cannot explain its superior therapeutic impact compared to glucocorticoids on muscle function. Actually, RvD2 and glucocorticoids have very similar impacts on fibrosis, inflammation, and necrosis. The only notable difference between these drugs is their effect on myogenic cells. While prednisone reduce the reserve of myogenic cells (Fig 2A,F and 7C), RvD2 increases the myogenic cell pool and myogenesis capacity (Fig 2F,G and 5C-F, 7C, and Suppl. Fig. S10). Overall, considering the low expression of Gpr18 in FAPs, the lack of direct impact of RvD2 on FAPs *in vitro*, and the non-superior effect of RvD2 on fibrosis *in vivo* compared to prednisone, we decided to focus on the impact of RvD2 on myogenic cells. We cannot completely exclude that RvD2 plays a potential role on FAPs behavior (which has been highlighted in the discussion of the manuscript); however, it will require a full characterization of FAPs that goes beyond the scope of this project (especially considering that by focusing mainly on myogenic cells and myogenesis we already have 8 main figures, and 12 supplemental figures for a total of 139 figure panels). We are currently exploring the role of Resolvins and other bioactive lipids on FAPs as part of a follow-up project (including transcriptomics, *in vitro* cell culture, genetically modified mouse, and different models of fibrosis).

Supplemental Figure S5. Gpr18 expression in different cell types in skeletal muscle. Co-immunofluorescence of Gpr18 (RvD2 receptor) and markers for different cell types was performed on *tibialis anterior* muscles of *mdx* mice. **A**) Representative pictures of F4/80 (macrophage marker; red), Gpr18 (green), and DAPI (blue). Arrowheads show double positive cells (F4/80⁺ Gpr18⁺; M1 macrophages) and arrows identify F4/80⁺ Gpr18⁻ cells (M2 macrophages). **B**) Representative pictures of Myog (a differentiated myoblast marker; red), Gpr18 (green), and DAPI (blue). Arrowheads show double positive cells (Myog⁺ Gpr18⁺). **C**) Representative pictures of CD25 (Treg marker; red), Gpr18 (green), and DAPI (blue). Virtually all CD25 cells are negative for Gpr18 as identified by white arrows. **D**) Representative pictures of Myog (orange), CD25 (red), Gpr18 (green), and DAPI, showing that Myog⁺ cells express Gpr18 (arrowheads) but not CD25⁺ Tregs (arrows). **E,F**) Representative Western blots and quantification of the

expression of Gpr18 on proliferative myoblasts and FAPs isolated from *mdx* mice and cultured *in vitro*. **G)** Growth curves of FAPs treated with RvD2 (200nM) or vehicle. **F,G)** Data are presented as mean \pm SEM; n=3-4 biologically independent samples performed in technical duplicates; * p <0.05.

- RvD2 does not affect MuSC expansion but stimulates their differentiation. Does the gpr18 expression varies with myogenesis (for instance no or low expression in myoblasts would explain no effect on their proliferation).

- We assessed Gpr18 expression during myogenesis as suggested by the reviewer. Western blot analysis of proliferating vs differentiated myoblasts indicates that Gpr18 is expressed at low level in proliferating myoblasts and is increased during differentiation (Fig. 3A,B, attached below). Immunostaining confirmed that Gpr18 is expressed in Myog⁺ cells (Suppl. Fig. S5B,D). Therefore, the preferential impact of RvD2 on myoblast differentiation and fusion could be mediated by the higher expression of Gpr18 in late myogenic stages. The lower expression of Gpr18 in muscle stem cells could also help to protect these cells against exhaustion, as confirmed by our *in vivo* data showing that the number of Pax7⁺ is unaffected by RvD2 administration, even at long term (Fig. 7A-D, see next point).

Figure 3A,B: **A)** Representative western blot and **B)** quantification of Gpr18 expression in proliferating myoblasts and differentiated myoblasts (1 day in differentiating medium) isolated from *mdx* mice. Data are presented as mean \pm SEM; n=5 biologically independent samples; ** p <0.01.

- Because the paper is written towards therapeutics, fibrosis should be analyzed in the severe model, or in diaphragm of *mdx* treated with RvD2. Also, a later time point than 21 days, (e.g. 2 or 3 months). Is the effect of RvD2 reversible?

- We fully agree with the reviewer's comments regarding the importance of fibrosis and long-term treatment in a therapeutic perspective. As recommended, we assessed fibrosis by Sirius red staining on the diaphragm muscle of *mdx* mice at short (1 and 3 weeks; Fig 5B,M,N, attached below) and long-term (2 months; Fig. 7B,F, attached below). Our results showed that at every condition tested, RvD2 significantly reduces fibrosis (to a similar level than prednisone). Long term experiments indicated that the effect of RvD2 is not reduced overtime. At 2 months of treatment, we still observe an increase in Myog⁺ cells in RvD2-treated *mdx* mice (without any reduction in Pax7⁺ MuSC pool), an increase in myonuclei per fiber, and an enhanced muscle function *in vivo* and *ex vivo* (Fig 7, attached

below). It was not possible to perform longer time points due to time constraint (especially in time of Covid-19, where our mouse colonies have been reduced and there was a constant risk of shutdown). However, these data show that there is no signs of reduced efficacy of RvD2 after 2 months of treatment, which suggest that its beneficial effects could be maintained for even longer period

Figure 7. The superior therapeutic effect of RvD2 is maintained at long-term in *mdx* mice. A) Timeline of weekly *ip* injection of either RvD2 (5 μ g/kg), prednisone (2 mg/kg) or vehicle into *mdx* mice for 2 months. B) Representative images of Pax7, Myog, Laminin, Collagen (Sirius Red) immunostaining in muscle section of *tibialis anterior* muscle (for Pax7, Myog, Laminin) or diaphragm muscle (for collagen staining) of treated *mdx* mice. Scale bars = 100 μ m. C, D) Quantification of total number of Pax7⁺ MuSC (C) and Myog⁺ differentiated myoblasts (D) in TA muscles of *mdx* mice treated for 2 months. E) Number of myonuclei per fiber in TA muscles of *mdx* mice treated for 2 months. F) Quantification of Sirius Red positive area (collagen) in diaphragm muscles of *mdx* mice treated for 2 months. G) Grip strength measured in treated *mdx* mice after 7 weeks of treatment. H) Specific muscle force.

force (N/cm²) of EDL muscles of *mdx* mice treated for 2 months. Data are presented as mean ± SEM; n=4 mice per group; C-G) *p < 0.05; **p < 0.01. H) *p < 0.05 compared with vehicle; † p < 0.05 compared with prednisone.

Minor points

1) Introduction lines 61 to 63. "MuSC display globally impaired myogenesis capacity". This should be rephrased since no report of intrinsic defect of myogenesis has been reported in DMD cells yet, except for the self-renewal, in a previous study of the authors.

Ref 12: the paper by Blau in 1983 is made on non-purified cell preparation.

Ref 13: if not mistaken, this paper does not report a deficiency in DMD cells but suggests a role of CD82 in myogenesis.

Ref14: in this paper we may consider that the cells are in a specific environment

Ref15 does not compare *mdx* and WT (except for oxidative stress) in the myogenic capacity of MuSCs.

Therefore while there is clearly a defect in myogenesis in *mdx*, this is not demonstrated that this defect is intrinsic to the cells yet (except for the self-renewal).

- We understand the concerns raised by the reviewers. We modified the text to put these references in their context. It is now indicated: "*Accumulating evidence indicate that the myogenesis capacity of dystrophin-deficient MuSC is impaired in DMD. Acute injury to dystrophic muscles leads to a deficit in MuSC activation and delayed/incomplete regeneration (20-22) and in vitro characterization of myogenic cells revealed a reduction in cell proliferation, differentiation and fusion (23-26). While the contribution of extrinsic factors (e.g. chronic inflammation) vs intrinsic mechanisms to these defects remains to be clearly defined, our previous work indicated that lack of dystrophin in MuSC intrinsically affect their capacity to perform asymmetric cell division, a process during which one MuSC generates one self-renewing MuSC and one committed myoblast (27, 28).*"

2) Figure 1: the cartoon in A shows the isolation of macrophages from monocytes. But the text indicates BMDMs.

- We collected cells from bone marrow and then we purified these cells by MACS to obtain a pure population of monocytes using the monocyte isolation kit (Miltenyi Biotec). Therefore, considering that macrophages are derived from purified monocytes, the term BMDM was removed from the manuscript.

3) How the concentration of RvD2 was chosen for in vitro experiments? Is there a dose-dependent effect of RvD2? How the concentration for the in vivo experiments was chosen?

- To determine the concentration of RvD2 *in vitro*, we first looked at what has been tested in the literature, which range between 10 nM to 1,000 nM (Keyes *KT et al*, Am J Physiol Heart Circ Physiol 2010, PMID: 20435846; Benabdoun HA *et al*, Arthritis Res Ther 2019, PMID: 30867044; Sawada Y *et al*, Scientific Rep 2018, PMID: 30089836; Duffield JS *et*

al, J Immunol 2006, PMID: 17056514). Thereafter, we tested different concentrations within that range (100nM, 200nM, 500nM) on primary myoblasts *in vitro*. Maximal effect on cell differentiation was reached at 200nM. The concentration for *in vivo* experiments was based on what has been described in the literature, which range from 2 µg/kg to 5 µg/kg (Giannakis N *et al*, Nat Immunol 2019: PMID: 30936495; Viola JR *et al*, Circ Res 2016, PMID: 27531933; Hellmann J *et al*, FASEB J 2011, PMID: 21478260; Xie W *et al* Lab Invest 2013, PMID: 23857007). We used 5 µg/kg and validated the effect by measuring the impact on inflammatory cell count in skeletal muscles.

4) Sup Fig1 versus Figure 2H. SupFig1 shows no difference of myogenic cell fusion with medium containing RvD2 (unless it is degraded in the medium in 48h) while RvD2 has a stimulating effect of myogenin expression. What is the difference between the 2 culture set ups? This is not clear.

- We hypothesized that RvD2 was degraded in the medium after 48h incubation. Indeed, different publications showed that resolvins have a relatively short half-life of a few hours (Krashia P *et al*, 2019 Nat Comm, PMID: 31477726; Giannakis N *et al* 2019 Nat Immunol: PMID: 30936495). To confirm the degradation of RvD2, we analyzed by ELISA the concentration of RvD2 in the medium after 48h of incubation. We observed that the concentration of RvD2 at 48h is approximately 10% of the initial concentration (200nM)(Suppl. Fig. S3A, attached below). Therefore, the remaining levels RvD2 in macrophage conditioned medium are too low to induce a significant direct impact on myogenic cells (Suppl. Fig. S3B).

Supplemental Figure S3. Myogenic effects of macrophage-conditioned medium are not mediated by remaining Resolvin-D2 in the medium. **A)** Concentration of Resolvin-D2 (RvD2) was assessed by ELISA in the culture medium 48h after the supplementation of RvD2. The final concentration was expressed as percentage on initial concentration (200nM). **B)** To determine if the low remaining level of RvD2 in the medium after 48 h of treatment affects myogenesis, we added RvD2 in the medium in a macrophage-free well for 48 h. The medium was added to differentiating myoblasts. Data showed no difference compared to control, indicating that the myogenic effect of the macrophage-conditioned media is driven by paracrine factors secreted by macrophages upon RvD2 treatment and not by remaining RvD2 in the medium. Data are presented as mean ± SEM; n=3-4 biologically independent samples.

5) Figure1B is unreadable.

- We separated the channel to clarify the figure (now Suppl. Fig. S2B). We also added other M2 macrophage markers in addition to CD206 (CD163 and Arginase-1) to further characterize the phenotype of these macrophages (Fig.1B,C).

6) Figure 3. The fusion index is very different than that shown in Figure 2H. What does explain the difference ?

- The only difference between these two experimental setups is that the cells were transfected with siRNA in Fig 3. It is likely that the transfection procedure have stressed the cells, which might have delayed fusion. However, since all the groups were treated equally in Fig 3 (all cells were transfected with siRNA scramble or siGpr18), we are very confident that these results are valid.

7) Please clarify the difference between Figure 6F and 6G. Is 6G the specific force? (so Y axis is wrong).

- Yes, it was the specific force. We are sorry for the confusion and we have changed the title of the Y axis accordingly (Fig. 6G)

Reviewers' Comments:

Reviewer #1:

Remarks to the Author:

The authors have done an excellent job and comprehensively addressed the reviewers' comments, and as a result the manuscript is substantially strengthened. My only final comment is to tone down the 3 sentences below, as, while the authors show that Grp18 mediates the RvD2 effect on myogenic cells, a formal demonstration that RvD2 binds Grp18 on myogenic cells is not provided.

- Abstract: "Moreover, Resolvin-D2 directly targeted myogenic cells by binding to the Gpr18 receptor, which promoted their differentiation and the expansion of the myogenic progenitor cell pool leading to increased myogenesis."

- Introduction: "1) by binding directly to the Gpr18 receptor on myogenic cells and activating the Akt-1 pathway".

- Discussion: "RvD2 mediates its effect by binding to the receptor Gpr18 on myogenic cells and activating the Akt pathway, which is known to promote myoblast differentiation and fusion".

Reviewer #2:

Remarks to the Author:

The authors strongly improved the quality and the strenght of this study, addressing all the concers raised by this reviewer.

Reviewer #3:

Remarks to the Author:

Thanks to the authors for having raised all my concerns, and for the very good experimental work that was done.

REVIEWERS' COMMENTS

We would first like to thank the reviewers and the editorial staff for their feedback that strongly improved the quality of this manuscript. Please find attached a point-by-point response to the few remaining comments.

Reviewer #1 (Remarks to the Author):

The authors have done an excellent job and comprehensively addressed the reviewers' comments, and as a result the manuscript is substantially strengthened. My only final comment is to tone down the 3 sentences below, as, while the authors show that Grp18 mediates the RvD2 effect on myogenic cells, a formal demonstration that RvD2 binds Grp18 on myogenic cells is not provided.

We agree with the reviewers and we have modified these sentences accordingly.

- Abstract: "Moreover, Resolvin-D2 directly targeted myogenic cells by binding to the Gpr18 receptor, which promoted their differentiation and the expansion of the myogenic progenitor cell pool leading to increased myogenesis."

It is now stated: "Moreover, Resolvin-D2 directly targets myogenic cells and promotes their differentiation and the expansion of the myogenic progenitor cell pool leading to increased myogenesis. These effects are ablated when the receptor Gpr18 is knocked-out, knocked-down, or blocked by the pharmacological antagonist O-1918."

- Introduction: "1) by binding directly to the Gpr18 receptor on myogenic cells and activating the Akt-1 pathway".

It is now stated: "This pro-myogenic effect of RvD2 is mediated by the Gpr18 receptor expressed by myogenic cells."

- Discussion: "RvD2 mediates its effect by binding to the receptor Gpr18 on myogenic cells and activating the Akt pathway, which is known to promote myoblast differentiation and fusion".

It is now stated: "This effect is mediated by the receptor Gpr18 expressed by myogenic cells. RvD2 treatment activates the Akt pathway in myogenic cells, which is known to promote myoblast differentiation and fusion^{71,72}"

Reviewer #2 (Remarks to the Author):

The authors strongly improved the quality and the strenght of this study, addressing all the concers raised by this reviewer.

Many thanks to the reviewer.

Reviewer #3 (Remarks to the Author):

Thanks to the authors for having raised all my concerns, and for the very good experimental work that was done.

Many thanks to the reviewer.